# Gamma oscillatory complexity conveys behavioral information in hippocampal networks

Vincent Douchamps [1], Matteo di Volo[2,3], Alessandro Torcini[3,4], Demian Battaglia [1,5,6,7] ✉ & Romain Goutagny [1,7] ✉

The hippocampus and entorhinal cortex exhibit rich oscillatory patterns critical for cognitive functions. In the hippocampal region CA1, specific gamma-frequency oscillations, timed at different phases of the ongoing theta rhythm, are hypothesized to facilitate the integration of information from varied sources and contribute to distinct cognitive processes. Here, we show that gamma elements -a multidimensional characterization of transient gamma oscillatory episodes- occur at any frequency or phase relative to the ongoing theta rhythm across all CA1 layers in male mice. Despite their low power and stochastic-like nature, individual gamma elements still carry behavior-related information and computational modeling suggests that they reflect neuronal firing. Our findings challenge the idea of rigid gamma sub-bands, showing that behavior shapes ensembles of irregular gamma elements that evolve with learning and depend on hippocampal layers. Widespread gamma diversity, beyond randomness, may thus reflect complexity, likely functional but invisible to classic average-based analyses.

Coherent oscillations of neuronal activity are ubiquitous across brain spatial and temporal scales[1,2]. Oscillations at different frequencies have been associated with the formation of sensory or behavioral representations[3,4], in the temporal organization of complex codes[5] or in the flexible routing of information between neuronal populations[6]. The possible functional roles of oscillations have been particularly investigated in the hippocampal formation, where, in the CA1 area of the dorsal hippocampus, convergent inputs could be disambiguated by the interaction of gamma and theta oscillations: different gamma-frequency carriers, timed at different phases of the ongoing global theta oscillations, would mediate information from different sources[7,8]. Hence, slow gamma (gamma$_S$; 30–80 Hz) predominates in the CA1 stratum radiatum (rad, where the inputs from CA3 are

localized) mostly at the trough/descending phase of CA1 pyramidal layer theta. On the other hand, medium gamma (gamma$_M$; 60–120 Hz) predominates in the CA1 stratum lacunosum moleculare (l-m, where the inputs from the entorhinal cortex layer 3 are localized), preferentially at the peak of CA1 pyramidal layer theta[8]. According to this prevalent model, layer-specific gamma oscillations in CA1 would identify the temporal dynamics of the afferent inputs, mediating specific memory-related processes (encoding for gamma$_M$ *vs* retrieval for gamma$_S$[7]).

Such a model, appealing for its simplicity and the link it proposes between distinct functions and discrete gamma sub-bands, may however fail to capture fully the richness of CA1 theta-gamma interactions[9]. Recent studies investigating gamma oscillations at the theta cycle

[1]Université de Strasbourg, Laboratoire de Neurosciences Cognitives et Adaptatives (LNCA), CNRS, UMR 7364, Strasbourg, France. [2]Université Claude Bernard Lyon 1, Institut National de la Santé et de la Recherche Médicale, Stem Cell and Brain Research Institute, U1208 Bron, France. [3]CY Cergy Paris Université, Laboratoire de Physique Théorique et Modélisation (LPTM), CNRS, UMR 8089, 95302 Cergy-Pontoise, France. [4]CNR - Consiglio Nazionale delle Ricerche, Istituto dei Sistemi Complessi, via Madonna del Piano 10, 50019 Sesto Fiorentino, Italy. [5]Aix-Marseille Université, Institut de Neurosciences des Systèmes (INS), INSERM, UMR 1106 Marseille, France. [6]University of Strasbourg Institute for Advanced Studies (USIAS), Strasbourg, France. [7]These authors jointly supervised this work: Demian Battaglia, Romain Goutagny. ✉e-mail: dbattaglia@unistra.fr; goutagny@unistra.fr

timescale reveal indeed a more dynamic and diverse landscape of gamma oscillations, with a broader variety of possible associations between gamma frequencies, theta phase, and an anatomical layer of occurrence (see[10] for a recent review). Yet, these studies continued to yield a classification of hippocampal gamma into distinct sub-types, reporting a multiplicity of supposedly typical average theta-gamma patterns: from two[11] to three[8] or more[12,13] gamma sub-bands (but see[14,15] for a contradictory hypothesis). Here, we refrain from distinguishing sub-types, acknowledging that even stochastic-like oscillations with fluctuating frequency and irregular timing can self-organize to process information[16]. We therefore characterize in detail the properties of individual transient gamma events, without ignoring their broad and ubiquitous variability, which may be informative about behavior rather than merely noise.

## Results

### Theta-gamma diversity is present in every CA1 layer

To characterize theta-gamma diversity, we analyzed local field potentials (LFPs) simultaneously recorded in the dorsal hippocampal CA1 area using 16- or 32-channel silicon probe ($n = 5$ mice; Fig. 1a). For every channel, we spectrally decomposed the LFP into its main frequency components through an unsupervised algorithm (EEMD approach[17], Supplementary Fig. 1a). We then computed the Current Source Density (CSD) signals using composite gamma LFPs (Fig. 1a), that is, the sum of the components peaking within a broad gamma band (30–250 Hz; Supplementary Fig. 1b–d). Such an approach avoids any filtering within narrow gamma bands imposed a priori. An analogous procedure was used to construct a theta composite signal from the hippocampal fissure (4–12 Hz; Supplementary Fig 1d; fissure theta shows larger, more defined theta cycles than pyramidal-layer theta but with a 180° phase-shift). We then performed a time-frequency analysis to segment the gamma CSD signal into short epochs corresponding to individual theta cycles (Fig. 1b). For each segment, we characterized each of its local peaks in the gamma spectrogram (Fig. 1c) as a multidimensional vector (i.e., a gamma element) describing its amplitude, frequency and phase of occurrence relative to the coincident theta cycle (3 gamma features), as well as the amplitude, frequency and asymmetry of this theta cycle (3 theta features). We restricted the extraction to the four strongest amplitude gamma elements per theta cycle, obtaining thousands of elements per channel and mouse (see theta and gamma counts on Supplementary Fig. 2 and our associated Zenodo drive with all the gamma tables used in this manuscript[18]).

According to the dominant view of a theta-phase and frequency specificity of the gamma contents between hippocampal layers, we calculated the mean probability density function of these two features per layer across mice (Fig. 2a, b). Unexpectedly, a substantial overlap was observed between layers for both variables, although the l-m presented slightly more gamma$_M$ events as well as more phase-locking

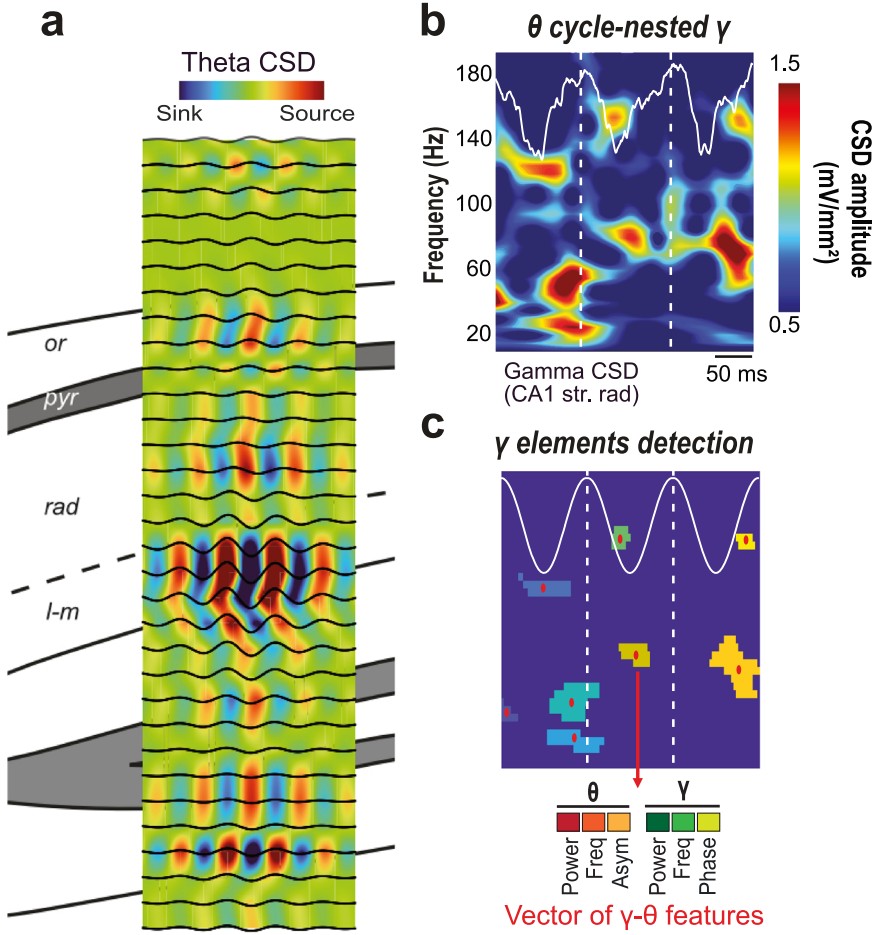

**Fig. 1 | Detecting and characterizing hippocampal gamma elements.**
**a** Electrodes along the silicon probe were localized in the different layers of the dorsal hippocampus using various indices to identify the hippocampal fissure, including the location of the maximum theta power and of the largest sink in the average theta-triggered CSD. **b** The gamma composite CSD wavelet spectrogram from each channel was first segmented into theta cycles (two consecutive peaks) from the theta composite signal recorded in the hippocampal fissure (white overlay). **c** local gamma peaks within the spectrogram were then detected within each theta cycle via a patch detection algorithm. These "gamma elements" were then characterized by extracting a vector of six features: three gamma features (amplitude, frequency, and theta-phase of the gamma element) and three theta features (amplitude, frequency and asymmetry of the coincident theta cycle).

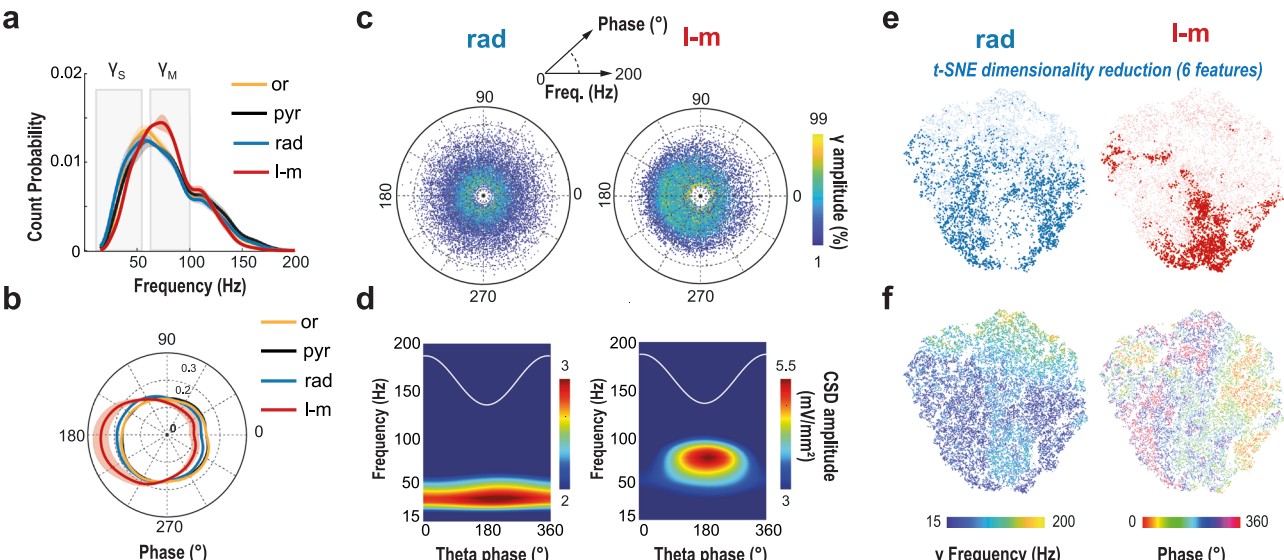

**Fig. 2 | Hippocampal CA1 layers have overlapping gamma frequency and phase distributions. a, b** Average distributions (mean pdf ± SEM; $n = 5$ mice) of gamma elements frequency and theta phase for each CA1 layer. Even if significant differences can be found between these distributions (see Supplementary Fig. 5 for details), whatever the layer considered, most of the gamma elements frequency distributions were spread across broad and overlapping ranges of frequencies, encompassing both the classical gamma$_S$ and gamma$_M$ sub-bands definition (full width at half maximum range: oriens (or), 35–97 Hz; pyramidale (pyr), 34–102 Hz; rad, 31–97 Hz; l-m, 38–100 Hz). **c** A joint representation of the three gamma features emphasizes the haphazard diversity of frequency (radius) and phase (angle) between gamma elements (dots) recorded in both the rad and l-m layers, especially at low amplitude (color: percentile of gamma amplitude). **d** Average theta-gamma spectrograms, on the contrary, put forward a marked distinction in frequency (and phase in a lesser extent) between the rad (left) and l-m layers (right), suggesting

these are respectively largely dominated by gamma$_S$ and gamma$_M$ oscillations. The apparent conflict between the representations in panels **c** and **d** is explained by the fact that average spectrograms are dominated by strong amplitude events. This is well visualized by dimensionally reduced representations (**e, f**) of the six-dimensional vectors describing gamma elements (obtained via a distance-respecting t-SNE algorithm). **e** Gamma elements from the rad and l-m layers cover similar areas in their joint bidimensional projection. However, the elements with high gamma amplitude (top 30%, dots with darker shade) occupy complementary zones for the two layers. **f** A color-coding by gamma frequency and phase of the same bidimensional projection shows that these strong elements tend to be: of gamma$_M$ type at theta trough, for the l-m layer; and gamma$_S$ at most phases, for the rad layer. These minorities of strong gamma elements are thus precisely the ones giving rise to rad and l−m average spectrogram peaks in panel (**e**). Panels **b**–**e**: examples from a representative mouse (mouse #3).

to the theta trough. We thus considered the joint distribution of the three gamma features for all the gamma elements per layer and animal (Fig. 2c for an example; Supplementary Fig. 2 for all mice and layers), with a similar conclusion: gamma elements were broadly scattered in both gamma frequency and theta-phase, although with a relatively stronger concentration of these in the l-m. Such diversity was confirmed even when extracting gamma elements with alternative techniques (e.g. filtering or independent-component analysis, Supplementary Fig. 3a) or from publicly available state-of-the-art recordings in rats[8] (Supplementary Fig. 3b).

However, when computing the rad and l-m average theta-gamma spectrograms using the same theta cycles than for the gamma elements characterization (Fig. 2d), we found that they were compatible with the classic, previously reported dichotomy between a gamma$_S$-dominated rad and a gamma$_M$-dominated l-m (Supplementary Fig. 2 for all mice and layers). In fact, our count approach revealed an increasing divergence between these layers in their frequency and theta-phase modes as the analysis was restricted to gamma elements with gradually stronger gamma power (see details statistics on Supplementary Fig. 4). The discrepancy between the two approaches (count vs average) thus indicates that average theta-gamma spectrograms are biased by only a minority of high-power transient gamma events.

This impression was confirmed by a dimensionality reduction analysis in which we projected in two dimensions the landscape of observed multi-dimensional gamma elements. We used a nonlinear t-distributed stochastic neighbor embedding (t-SNE[19]) that attempts preserving distance, so that elements close (or far) between them in the original six-dimensional source space were still close (or far) on the

two-dimensional plane even after their projection. All elements from all anatomical locations are projected simultaneously, however for convenience we can plot separately for different layers, using precisely the same coordinates, as we do in Fig. 2e for a representative mouse. Strikingly, the domains covered by the projection of rad and l-m elements were largely overlapping, at the exception of the ones with the highest power, highlighted in darker color and occupying clearly complementary zones. The same projection can be used to also represent the distributions across elements (now for both rad and l-m combined) of other gamma element features, as in a thematic atlas. Figure 2f shows a map of the frequencies and phases of different gamma elements (see Supplementary Fig. 5 for other features and mice), confirming once again the wide diversity of features and the lack of a simple way to discriminate between rad and l-m elements in terms of few features only.

We could also verify that this diversity originates from the local circuits generating the inputs received by CA1. To do so, we took advantage of recent publicly available recordings made in mice undergoing transient deafferentation of CA1 via silencing of the entorhinal cortex and/or CA3 inputs[20]. Suppression of either one of the inputs impacted the density of recorded gamma elements scattered at all frequencies and phases, confirming that diversity is already present in the suppressed inputs (see the difference maps of Supplementary Fig. 6). We remark, however, that EC input was more structured than CA3 input, as its suppression caused a more marked decrease of medium gamma elements with a stronger locking, corresponding to the high amplitude elements already highlighted in Fig. 2d (right). Yet, even for EC input, suppression effects were widespread (see *Discussion*).

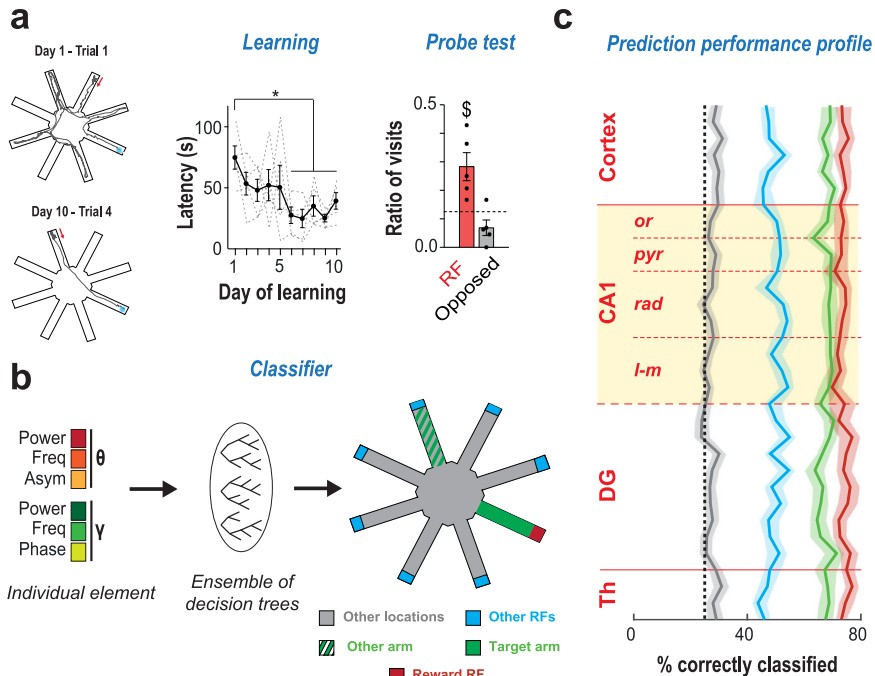

**Fig. 3 | Location during exploration behavior can be decoded from individual gamma elements. a** In our spatial navigation task, reward is located at the end-box of a fixed target arm in a radial maze. The mouse enters the maze from a different arm at every trial. Few trials are performed every day, over several days. Left: example trajectories at different days of learning. Middle: the latency to reward decreases across days ($n = 5$ mice; mean $\pm$ SEM; Mean $X^2_{(9,49)} = 19.342$, $p = 0.02$; Friedman ANOVA), indicating that mice learn the task. Right: In probe trials, no reward is given at the reward location. Mice spend a larger amount of time exploring the former rewarded arm than an opposite one, denoting memory of the reward location (one-tailed t-test, $t_{(4)} = 3.22$; $p = 0.032$). **b** We trained tree-ensemble classifiers to decode rough location within the maze (target arm, reward RF, other RFs and the remaining locations) from individual gamma elements (three theta and three gamma features, cf. Figure 1b). We also trained alternative classifiers to detect an alternative arm, remote from reward. **c** Fraction of correctly classified locations, by maze location (colors as in **b**), for a representative mouse (mouse #3; see Supplementary Fig. 7 for other mice and prediction performance with alternative feature sets and in probe trials). Different classifiers were trained for different depths along the dorsal hippocampal axis (cf. Fig. 1a). Solid lines indicate average performance across all trials (shading, 95% bootstrap c.i.). Performance in detecting target arm, reward and other arms RFs was significantly above chance level (dashed black line) for every anatomical layer.

In conclusion, we found no evidence for narrow gamma bands[15]. The actual observations are eventually more compliant with a description in terms of diverse ensembles of transient gamma oscillations, widely scattered in frequency, phase, and other features.

## Navigation behavior can be decoded from individual gamma elements

Is this diversity functional or just noise without meaning? In other words, does behavior shape the seemingly stochastic dynamics of gamma elements? To answer this question, we trained the mice to learn a novel spatial reference memory task (Fig. 3a). In this task, mice seek for an appetitive target located within an 8-arm radial maze. They need to learn a unique, stable goal location over multiple days of training (10 days; 4 daily trials). The change of departure arm in each of the few daily trials enforces the comparison of allocentric cues with internal representations. We therefore attempted decoding the current position of the mouse within the maze during navigation behavior, based on the features of the simultaneously recorded gamma elements. To do so, we trained machine learning classifiers (ensembles of randomized decision trees, Fig. 3b) to predict the rough location of the mouse (four non-overlapping maze sections: target arm approach, target arm reward field (Reward RF), other arms' reward fields (other RF, no reward) and rest of the maze; Fig. 3b) based on the six-dimensional vector parametrization of a coincident individual gamma element recorded on a specific channel. The training set for each classifier was restricted to elements from a subset of randomly chosen theta cycles, the unused elements being allotted for later cross-validation of the performance. Decoding yielded performances well

above chance-level, particularly for the target arm and the reward field, for any layer within CA1 (Fig. 3c for a representative mouse; Supplementary Fig. 7a–e for all mice). Given the limited modulation of the performance by the anatomical position (channel), we summarized the achieved channel-averaged performances of decoding for all mice. We showed that performance for decoding target arm, reward field, and other arm end-fields were well above the chance level for every mouse (Supplementary Fig. 8a, see also the confusion matrix in Supplementary Fig. 8b). Even if the location could be better decoded from events whose amplitude belonged to the largest quartile of the amplitude distributions –the one dominating spectrograms (cf. Figure 2d)–, decoding was possible even from elements at weaker amplitudes in the other quartiles, with the exclusion of only the lowest quartile of amplitudes (Fig. 4a for target arm and reward field decoding performance by gamma amplitude quartile; Supplementary Fig. 8c for other locations). As hippocampal theta and gamma oscillations are modulated by speed[21,22] we analyzed whether the performance of location decoding depends on the speed of movement of the mouse (Fig. 4b and Supplementary Fig. 8d). Decodability of reward field and other arms ending zones was higher when speed was low (lowest quartile) and when speed was high (highest quartile) for the target arm. When speed was large, decoding was significant for all four considered maze locations and confusion between locations was reduced (Supplementary Fig. 8d). Yet, we were able to significantly decode location from elements in other speed quartiles (down to the second quartile for target and up to the fourth quartile for reward) indicating that speed is not the unique determinant of gamma element modulations by maze location (note that speed distributions over the different maze

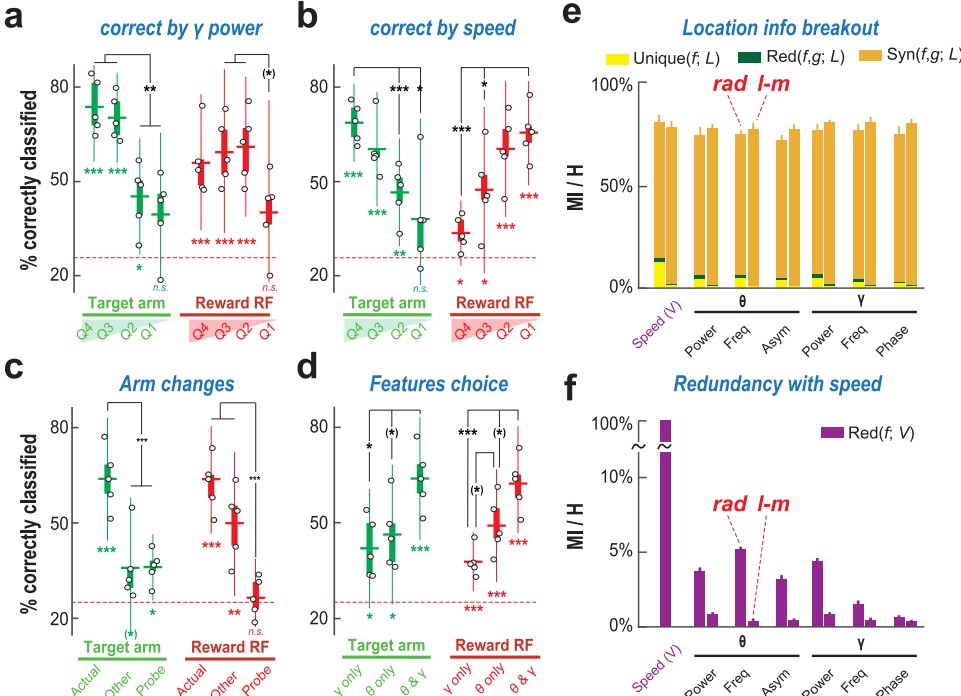

**Fig. 4 | Decoding performance is robust, genuine and synergistic. a** Dependence on gamma amplitude. Performance is significantly higher for amplitudes in larger than smaller distribution quartiles ($p < 0.002$ for target; $p < 0.033$ for reward), however it remains significant for all but the lowest quartile ($p < 0.006$ for target Q3). **b** Dependence on motion speed. Decoding performance for target arm (reward RF) was higher for larger (lower) quartiles of speed ($p < 0.004$ for target, $p < 0.001$ for reward), but was significant even for low (high) speed quartiles (target Q3, $p < 0.0025$; reward Q1, $p < 0.011$). **c** In probe trials, decoding performance dropped for both target arm ($p < 0.0005$) and reward RF ($p < 0.0001$). Decodability of a generic other arm was lower than for target arm ($p < 0.0027$), but performance did not drop significantly for reward. **d** When training classifiers to decode maze location based on reduced input feature sets (only gamma- or theta-related features) decoding performance dropped (e.g., when comparing gamma-only with combined theta and gamma inputs, $p < 0.0199$ for target arm and $p < 0.0014$ for reward RF). $n = 5$ mice; dots, performance averaged over trials and electrodes for different mice; boxes, IQRs and sample mean; whiskers, 95% sample c.i.; *, $p < 0.05$; **, $p < 0.01$; ***, $p < 0.001$ after Bonferroni correction; symbols in brackets denote significance only before Bonferroni correction; one-tailed t-tests of sample vs chance level; two-tailed t-tests between samples.) **e** Fractions of maze location information conveyed by pair of features were large (bars, averages over mice and feature pairs, grouped by pairs including a specific feature –i.e., $n = 15$ pairs per feature time $n = 5$ mice–, for representative rad and l-m channels; whiskers, 95% bootstrap c.i.). Mutual information was mostly due to synergy between features, which conveyed little unique or redundant information about location. **f** Speed accounted for a small fraction only of the variability of individual gamma element features, as revealed by normalized mutual information with speed ($n = 5$ mice, averages over all mice and features, for representative channels; whiskers, 95% c.i estimated from twice sample standard deviation). We don't show individual mice values in panels (**e**) and (**f**) to avoid figure crowding.

sections are not identical but, still, largely overlapping, cf. Supplementary Fig. 8l). Analogously, decodability was maintained across several quartiles of the other features in the gamma element parametrization, that is, gamma frequency and phase relative to theta, and theta cycle amplitude, frequency and asymmetry (Supplementary Fig. 8e–i). In short, decodability was not limited to narrow categories of elements with specific feature combinations, but was on the contrary rather pervasive, extending notably to gamma elements strongly deviating from spectrogram averages. In some cases, more information could even be extracted from weak than from strong amplitude gamma transients (as when decoding presence in the reward-less other RF locations, Supplementary Fig. 8c).

To further verify that our classifier extracted genuine behavior-related information from individual gamma elements, we first modified our classifier design by training the classifier not to specially identify the target arm but a randomly chosen arm among the behaviorally non-saliant (i.e., neither departure nor target arm) ones. The decoding performance that could be reached for these generic arms was not as high as when decoding the approach to the actual target arm (although the decoding of the reward zone was not significantly changed, Fig. 4c, and Supplementary Fig. 8j "other" boxes). This indicates that classifiers can detect signatures in gamma elements –akin to an "eureka" signal– which specifically reflect behaviors observed when approaching the target arm but no other arms of the maze. Second, at

the end of task learning, we performed a probe trial in which the reward was removed. Such probe condition modified the behavior during target arm approach and reward field exploration (cf. Figure 3a, right), as reward was unexpectedly missing at the previously learned location and context was thus altered. As shown by Fig. 4c and Supplementary Fig. 8j ("probe" boxes), classifiers trained to decode target arm and reward fields in learning trials could still significantly decode transit in the target arm zone (although with a lower performance) but the performance in decoding the reward field dropped at chance level. Such pattern of performance modification was consistently observed across all recording channels and mice (cf. Supplementary Fig. 7f–j). Thus, behavior induced by the probe condition translates into modified gamma element signatures, since the same classifiers that decoded relevant maze locations in preceding trials could not identify them anymore in the probe trial. Therefore, the features of individual gamma elements –very diverse, especially when gamma amplitude is low– are modulated by maze location and behavior in complex but consistent ways that machine learning classifiers can successfully identify.

## Different features of theta and gamma oscillations synergistically reflect behavior

Which features give the largest contribution to the successful decoding of maze location from diverse gamma elements? To address this

question, we constructed machine learning classifiers using, as input alternative, smaller subsets of features: only the three gamma features (gamma-only) or only the three theta features (theta-only). Target arm and reward field could still be decoded above chance level based on the gamma-only or the theta-only subset of features, however, the performance dropped with respect to the original classifier, indicating that theta and gamma-related dimensions of the gamma elements convey non-redundant information (Fig. 4d and Supplementary Fig. 8k). Information theory can be used to further investigate the contribution of different features to classifier performance. Indeed, there are various ways in which the joint consideration of multiple input features can yield more information about the target output. First, each of the input features may convey some information that none of the other input features convey, so that combining more of these unique information contributions yields more overall information. Second, some features may convey the same information, but corrupted by independent noise, and this redundancy may be helpful to achieve a better signal-to-noise ratio. Third, and perhaps more interestingly, combinations of features may convey synergistic information "beyond the sum of the parts", which cannot be accessed when considering features independently. We focus here on considering mutual information between $L$, the target output maze location, and pairs of possible inputs $f$ and $g$ (e.g., theta and gamma amplitudes), since, in this case, the Partial Information Decomposition (PID[23]) framework provide precise guidelines for decomposing this total mutual information into its unique, redundant and synergistic fractions.

We first found that, on average, pairs of gamma and theta feature carried over 75% of the information needed to perfectly specify maze location at any time, thus explaining why decoding of location is feasible. This is revealed by the total height of the vertical bars in Fig. 4e, where we show the average mutual information between maze location and pairs of input features, pooled by included feature (i.e., all pairs including gamma frequency, all pairs including gamma amplitude, etc.; see also Supplementary Fig. 9 for detail on individual non-averaged pairs). Next, we fractioned the total mutual information into the parts constituted by the unique, redundant, and synergistic fractions (respectively in yellow, green and orange colors, Fig. 4e). Remarkably, the synergistic fraction was by far the most important, accounting in most cases for over 70% of the total information conveyed by the feature pairs about maze location. These indicate that individual theta- and gamma-related oscillatory aspects have individually complex and changing relations with maze location (hence the low unique information fractions) but that their joint patterns of general covariation do depend on it (hence the large synergistic fractions).

We also considered mutual information between maze location and speed of movement, as the distributions of the speed of movement were not completely identical for different maze sections (Supplementary Fig. 8l). As shown by the leftmost bar of Fig. 4e, predictor pairs including speed among the input variables did not convey significantly more total information about maze location than any other pair of gamma element features. Yet, even if feature pairs involving speed do not carry more maze location information, the encoding of maze location by oscillatory features may still indirectly reflect relations with speed, via the dependence of the oscillatory features themselves on speed. Therefore, we also computed the redundancy of speed with the other oscillatory features. Individual oscillatory features of the rad shared more information with speed than oscillatory features of the l-m (see Supplementary Fig. 9). However, the shared information with speed never explained more than 5% of their variation entropy (Fig. 4f). Together these results indicate that variations of gamma element features are not completely explained by speed, but synergistically convey genuine maze location information, beyond mere speed variations across locations. The dramatic drop in inter-feature synergies observed during probe trials may thus explain the

lower maze location decoding performance in these with respect to learning trials (Fig. 4c and Supplementary Fig. 8j and 9).

## Complex gamma ensembles evolve with task learning

Behavior can be decoded out of gamma elements, but is the decoding grammar similar across learning? And is the decoding performance improving with training? We explored this by training classifiers over gamma elements from trials within restricted ranges, starting from early trials and then sliding the inclusion range to the latest trials. Gamma element outstanding diversity was present at any trial range, noticeably never losing their continuous and broadly dispersed frequency and theta-phase distributions despite slight changes (Fig. 5a, see also the t-SNE projections in Supplementary Fig. 6c and polar plots in Supplementary Fig. 10). Maze location information could be significantly decoded from these diverse gamma elements at any trial range, although the detailed profiles of variation across learning were heterogeneous for different mice, possibly reflecting idiosyncratic navigation learning strategy. Yet, the cross-validated fraction of correct predictions was larger for late than for early trials, with a performance improvement on average of -7% for the l-m and of -15% for the rad layers (Fig. 5b).

We then compared the complex ways in which gamma element variations reflected maze location by adopting a cross-classification approach. Classifiers trained on trials within a specific training trial range were used to predict maze location on trials from another testing trial range, and the obtained fractions of correct prediction were compiled into cross-prediction performance matrices (see Fig. 5c for a representative rad layer example and Supplementary Fig. 11a for rad and l-m layers in all mice; cross-validation cannot be used in cross-classification, so we plot direct cross-prediction errors, hence slightly different performance ranges, see *Methods* for a detailed explanation). The obtained matrices of cross-prediction performance across trial ranges were characteristically asymmetric: the larger upper than lower triangular parts indicate that a classifier trained in one trial range can better predict location from past rather than future trial ranges (cf. more yellow above the diagonal in Fig. 5c and Supplementary Fig. 11a; see also Fig. 5d for a quantification). The performance of decoding yet dropped when the training and testing trial ranges were separated by a timespan too large (cf. blue zone at the upper right corner in Fig. 5c and Supplementary Fig. 11a). We interpret these findings as an indication that the complex mapping of maze location by gamma ensemble features is not frozen but smoothly evolves through time. However, the drift of this mapping is specifically shaped by previous experience, hence the existence of an "arrow of time" in cross-trial decodability.

## Complex gamma ensembles are spatially organized

Frequency, phase, and other properties of gamma elements are nearly equally distributed across CA1 layers. Can we still find differences among anatomical locations despite this apparent homogeneity? A possibility is that multiple gamma ensembles co-exists and, despite their comparable spectral variability, display different associations with behavior, reflecting distinct functional roles. To probe this hypothesis, we adopted once again a cross-classification approach to compare the mappings of maze location by gamma ensembles recorded at different anatomical locations. Figure 6a shows a representative matrix of the fraction of correct predictions obtained when training a classifier on gamma elements recorded on a channel and testing it from gamma elements recorded on another channel (see Supplementary Fig. 11b for all mice). This matrix displays a hierarchical block organization. Classifiers trained on channels within the hippocampus can decode maze location from other hippocampal, but not extra-hippocampal, channels (and vice versa). Furthermore, within CA1, at least two blocks can be distinguished including channels located within the pyr and upper rad layers, and lower rad and l-m layers,

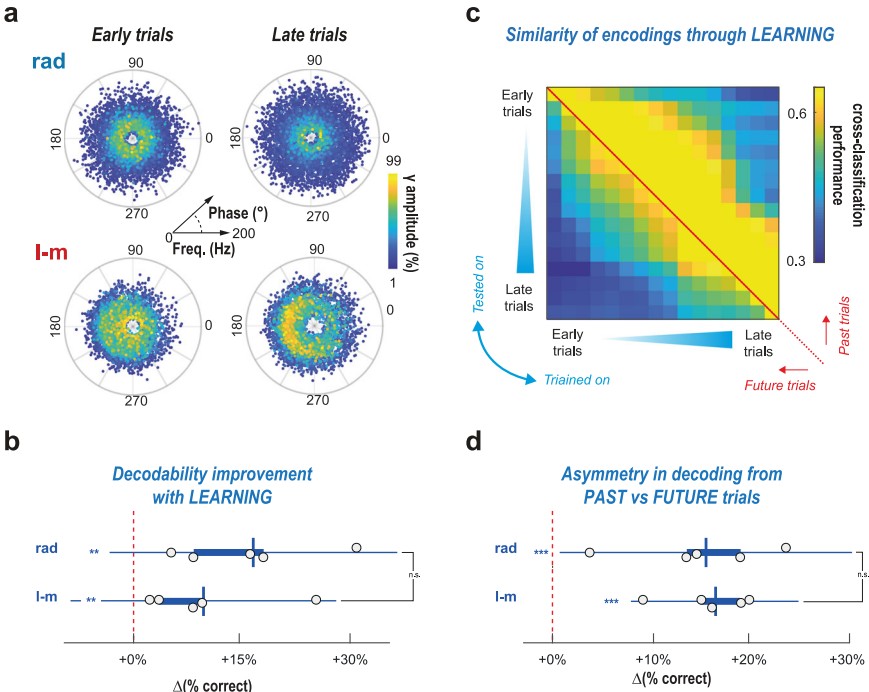

**Fig. 5 | Relations of gamma elements with behavior depend on training. a** Polar scatter plots of individual gamma elements distribution (as in Fig. 1d), separate for early (days 1–3) and late (days 8–10) trials (*n* = 5 mice). Wide diversity of gamma elements features is observed at all stages of task learning (here for mouse #3; see Supplementary Figs. 2, 5a and 10 for more details and all mice). Although remaining complex, the distributions are however evolving with learning. **b** The average performance of decoding maze location (average over all classes, for reference channels in rad and l-m) is higher for late than for early trials, as revealed by boxplots of percent performance improvement for representative channels in both rad and l-m layers (*p* < 0.013 for rad and *p* < 0.04 for l-m). **c, d** We performed cross-classification analyses, training classifiers to decode maze location from gamma ensembles in a range of trials and using them to extract information from other trials in past or future time ranges. **c** The resulting cross-prediction error matrix (here for a representative rad channel for mouse #3; see Supplementary Fig. 11a for l-m layer and other mice) is asymmetric with respect to the diagonal, indicating that classifiers trained on future trial ranges can decode information from past trial ranges better than in the opposite direction. **d** This asymmetry is quantitively confirmed by positive percent difference between performances in past-on-future or future-on-past prediction directions (positivity of the increment, *p* < 0.006 for rad and *p* < 0.0006 for l-m). In the boxplots of all this figure's panels dots denote performance improvements for different mice. boxes, IQRs; horizontal line, sample mean; whiskers, 95% sample c.i.; **\**p* < 0.01; **\*\**p* < 0.001 after Bonferroni correction. One-tailed t-tests are used for both comparisons of samples with chance level; and between samples.

respectively. Cross-decodability between classifiers was high between channels from the same block, but low with extra-block channels, indicating that at least two types of CA1 gamma ensembles exist, differentially modulated by behavior despite their large overlap in frequency and phase distributions.

We then repeated this spatial cross-classification analysis but separately for earlier and later trials along the learning of the task (Supplementary Fig. 11b). We found that the cross-decodability between the l-m-like and upper rad-like channel blocks increased in later trials. In general, cross-decodability increased with task learning between all channels. However, this was particularly noticeable for classifiers trained within the l-m-like channel block as they gradually improved in decoding the maze location from gamma elements recorded in the upper rad-including channel range (Fig. 6b). Such results suggest a convergence of current sensory representations conveyed by entorhinal inputs to the l-m layer, onto internal model representations, provided by CA3 inputs to the rad layer, suggestive of potential prior learning[24] (see *Discussion*).

**Sparse firing at the "fringe-of-synchrony" underlies complex gamma ensembles**
We have shown that hippocampal gamma elements convey genuine behavioral information, evolve with learning, and are related to anatomy. But what could be the circuit-level mechanisms generating this diversity and its relation to behavior? We hypothesize that this diversity stems from the dynamic regime of operation of the local networks (in entorhinal cortex or CA3) generating input signals to CA1. Such

diversity, natural consequence of randomness in recurrent synaptic connections combined with balanced excitation and inhibition, would then be shaped by the detailed firing of excitatory and inhibitory neurons in the source population. The specific amplitude, frequency, and phase of individual gamma elements observed within CA1 would be modulated by the detailed firing patterns in the input regions, essentially representing a blurred reflection of the active neuronal assemblies at a given moment. In this view, the behavioral information conveyed by individual gamma elements, even those with weak amplitudes, would be an indirect representation of the underlying information carried by the firing patterns in the source regions. To corroborate this hypothesis, we constructed a simple spiking model for balanced excitatory-inhibitory populations with random recurrent connectivity, representing a generic gamma-generating input source to CA1.

Specifically, we considered a network with thousands of randomly interconnected excitatory (E) and inhibitory (I) quadratic integrate-and-fire (QIF) neurons[25], driven by an external theta-modulated current input and we simulated unit activity and the associated LFP-like signals (Fig. 7a). Figure 7b shows a representative raster plot of activity and Fig. 7c the corresponding LFP spectrogram. As a first result, we found that gamma diversity, rather than surprising, should be expected, as it robustly emerges for most parameter combinations in the spiking model, provided the network remains not too far, but still below a transition to strongly synchronized oscillatory firing (i.e. at the "fringe of synchrony", rather than in a regime with fully developed synchrony). Extracting gamma elements from simulated gamma

# a

**Similarity of encodings through REGIONS**

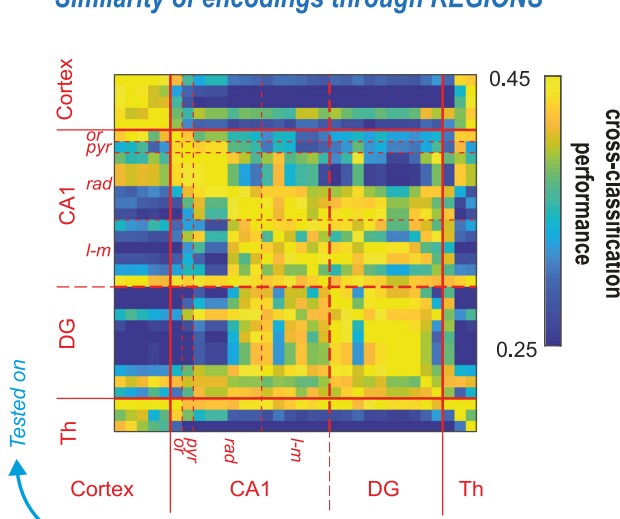

# b

**Cross-decodability improvement with LEARNING**

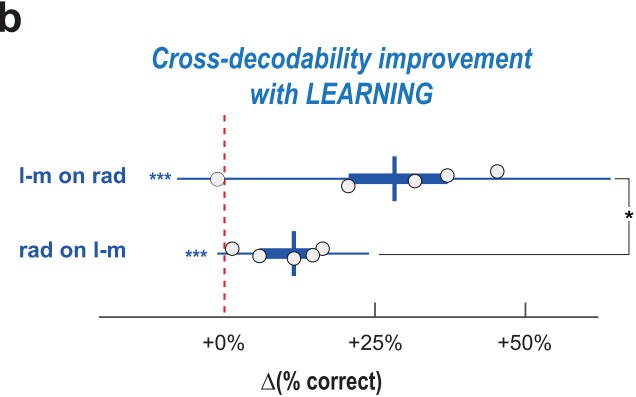

**Fig. 6 | Relations of gamma elements with behavior depend on and anatomical layer.** We studied cross-classification between different recording locations and its variations along task learning. **a** The cross-prediction error matrix (all trials, mouse #3; see Supplementary Fig. 11b for other mice) displays a block structure, indicating that different anatomical locations have alternative types of gamma elements to behavior inter-relations. Hippocampal and non-hippocampal channels form different blocks. Within hippocampus CA1, superior rad and l-m channels belong as well to different sub-blocks. **b** These anatomically-organized patterns of inter-relations evolve along task learning, as revealed by increased l-m vs rad cross-decodability in late with respect to early stages (percent improvement of cross-prediction performance; $p < 0.016$ for both l-m-on-rad and rad-on-l-m cross-prediction directions). The improvement in cross-predictability across learning was larger in the l-m-to-rad than in the rad-to-l-m direction (significance of difference, $p < 0.042$). In the boxplots boxes denote IQRs; horizontal line, sample mean; whiskers, 95% sample c.i.; *, $p < 0.05$; **, $p < 0.01$; ***, $p < 0.001$ after Bonferroni correction. One-tailed t-tests are used for comparisons of: samples with chance; and between samples.

spectrograms yields a diversity of gamma frequencies and phases comparable to actual data (cf. radar plots in Fig. 5a and Fig. 2c). The shown simulation is obtained for a specific choice of parameters (square working point in Fig. 7d), however, the displayed gamma diversity is preserved over a very broad range of conductance and external drive intensity. This is quantified by large spectral entropy values in Fig. 7d, remaining uniformly high over most of the represented parameter space, colocalizing with regimes of weakly synchronized and low-amplitude oscillations (triangle, circle and square example working points in Fig. 7d and Supplementary Fig. 12a).

Spectral entropy drops uniquely above a transition to a strongly synchronized oscillatory regime, characterized by higher amplitude fluctuations of the mean membrane potential (star symbol in Fig. 7d). Wherever spectral entropy is high, oscillations are transient and display scattering in frequency and phase like in empirical data (four paradigmatic cases are characterized in Supplementary Fig. 12c). In simulated ensembles of gamma elements, as in real data, frequencies are more narrowly tuned in oscillatory events with high amplitude and the exact frequency at which these high amplitude events tend to occur can be smoothly controlled by varying the strength of the coupling of excitatory to inhibitory neurons (triangle circle, and square symbols in Supplementary Fig. 12b–c).

We also considered the regularity of single-neuron spike trains. Over the broad range of parameters associated with large gamma element variability, the average firing rate of neurons is small compared to the average frequency of gamma oscillations (Supplementary Fig. 13a), so neurons do not fire at every gamma cycle and their spike trains are temporally irregular (note however that in our model, firing rate of excitatory neurons is higher than the one recorded in the hippocampus or entorhinal cortex of behaving rodents Supplementary Fig. 13b, d) associated to large spike train entropy values (Fig. 7d). On the contrary, entering the high amplitude fluctuation regime (star symbol), spike train entropy drops, indicative of a regime of spike-to-spike synchrony, in which neurons tend to fire nearly at every gamma cycle (Fig. 7d). In such a regime, only a limited amount of information could be conveyed by spike patterns, as most neurons are active within each gamma cycle. On the contrary, in the high entropic regime, richly informative "codewords" could be constructed by monitoring which neurons are active and which are silent at each specific gamma cycle. In other words, regimes with high gamma element variability are expected to have a higher coding capacity.

Finally, to prove that gamma element variability is shaped by neuronal firing patterns, we trained a random forest classifier to predict from the vector parameterization of specific gamma elements the firing or silence of each given neuron in a small window centered on the input gamma element. We focused specifically on E neurons, whose firing was sparser (Fig. 7e). We then quantified the performance of decoding by evaluating the fraction of true and false positive output inferences. The performance was heterogeneous and for many neurons, the true positive fraction did not get above 30% (Supplementary Fig. 13c). However, for a subset of 24% of neurons, decoding was possible with a very accurate performance (Fig. 7f), with both true positive and true negative fractions above 70%, for a precision of ~72% (fraction of predicted spikes that are existing) and a recall of ~74% (fraction of existing spikes that are predicted). These "decodable" neurons did not have significant differences in firing rate, the strength of external drive or yet degree of local connectivity with respect to the other, "undecodable" neurons. However, their firing was more phase-concentrated relative to the ongoing theta oscillation (Supplementary Fig. 13d) and tended to occur in phase ranges immediately preceding or following theta oscillation peak (Supplementary Fig. 13e). Even if these details may not be generalizable beyond the specificities of our model, still we are able to prove that decoding the firing of a fraction of the population neurons from gamma elements is possible. Therefore, the variability of gamma element features reflects, at least in part, the underlying neuronal firing variability and could inherit from it the capacity to –indirectly– carry information about behavior.

## Discussion

Using machine learning-based decoding of electrophysiological recordings during a behavioral task, we showed that in vivo hippocampal theta-gamma oscillations are not well described by sharply distinct narrow-band modes. On the contrary, at every CA1 channel, we observed broad distributions of gamma frequency and theta-phase of appearance, largely overlapping between distinct anatomical layers

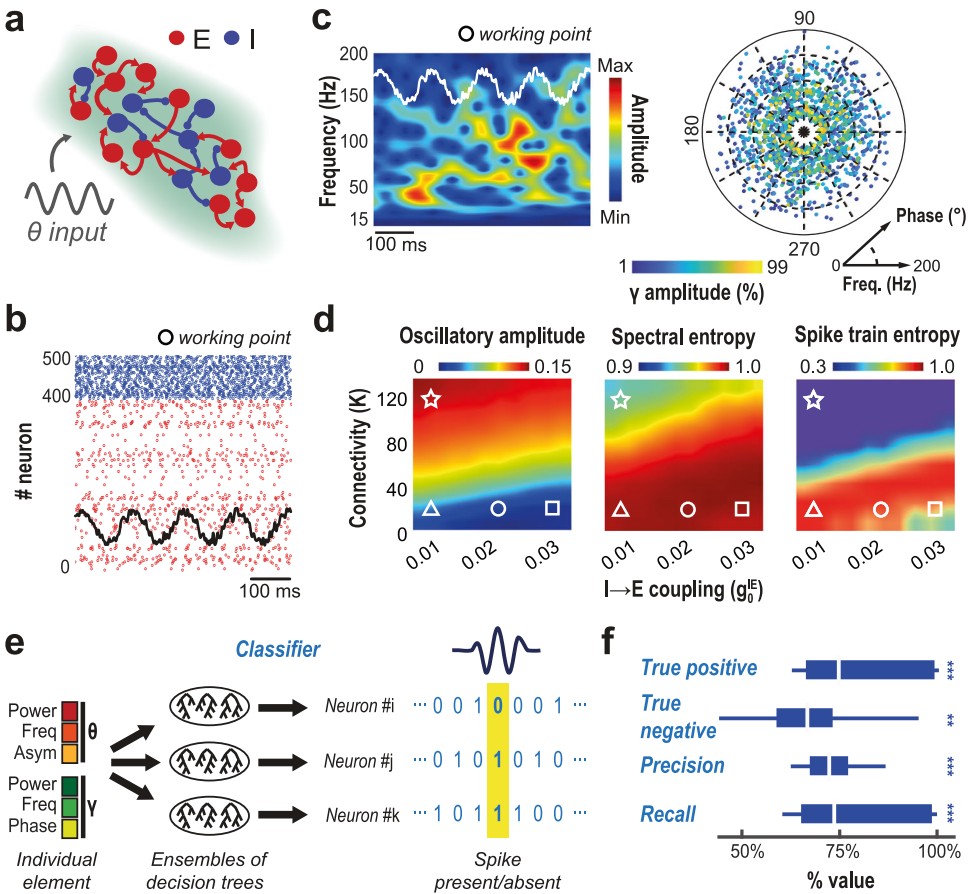

**Fig. 7 | Large diversity of gamma elements reflects firing patterns at the fringe-of-synchrony. a** We generated simulated LFP-like signals using a computational model of a generic local circuit, generating gamma oscillations and driven by an external theta-modulated input current. The model network included thousands of randomly interconnected spiking excitatory (E) and inhibitory (I) neurons. **b** Typical raster plot of the spiking activity of selected neurons, with superposed trace of the associated LFP-like signal computed from the model. **c** Spectrograms of the gamma composite component of simulated LFP-like signals reveal the existence of transient gamma oscillatory events at variable frequencies and phases with a landscape of gamma elements diversity comparable to real recordings. **d** The diversity and frequency distribution of simulated gamma elements depend on model parameters, such as the density of within-population connectivity K and the average strength of I-to-E synaptic coupling. The parameter-dependency surface of spectral entropy (middle panel) shows that narrower-band oscillations with a more precise frequency occur only when connectivity K is very large (star working point). However, in this case, the level of population synchronization would be unrealistically large (large signal standard deviation; rightmost panel). The degree of

synchronization also correlates with the average entropy of individual spike trains (quantifying their temporal irregularity). Entropy is large at the "fringe-of-synchrony" (triangle, circle and square points, corresponding to oscillations with different mean frequencies, cf. Supplementary Fig. 12), when spikes are emitted only every few gamma oscillatory cycles in a random-like fashion, while is low in the more synchronous regime (star point) where synchrony is spike-to-spike. **e** We constructed a classifier predicting whether a given neuron is emitting a spike or not within a small window centered on a gamma element, whose parameterization is fed as input to the classifier. **f,** For a subset of more precisely phased neurons (cf. Supplementary Fig. 13), successful decoding was possible, with precision and recall above 70%, indicating that gamma element diversity reflects spiking patterns (for all comparisons with chance level, at least $p < 0.023$ or smaller). Estimations of Entropy and decoding are performed for E neurons only, as they require binning of spike trains and bin-size have been optimized to E average firing rate. In the box-plot, boxes denote IQRs; horizontal line, sample mean; whiskers, 95% sample c.i.; **, $p < 0.01$; ***, $p < 0.001$ after Bonferroni correction. One-tailed t-tests are used for comparisons of samples with chance level.

(Fig. 2, Supplementary Figs. 2–5). This diversity of gamma oscillations is widespread and common to all analyzed datasets, including data previously used to conclude the existence of narrow gamma bands (Supplementary Fig. 3). Consequently, if we were given the phase and frequency of an individual transient gamma element without being told from which layer this element has been recorded, we would be generally unable to provide an answer to this question. This apparent homogeneity in oscillatory features, broken only for the strongest amplitude events, does not prevent the existence of distinctions between layers. These distinctions, however, are not anymore just of spectral nature, but instead emerge at an "algorithmic" level[26], i.e. in the way in which this pervasive gamma variability relates to behavior (Figs. 3, 4, Supplementary Figs. 7–9) and evolves through learning (Figs. 5, 6, Supplementary Figs. 10–11).

Hippocampal oscillations would thus be better described in terms of a collection of complex gamma ensembles, i.e. collections of

transient oscillatory events that, despite their heterogeneity in phase and frequency, are distinctively modulated by both behavior and learning. In other words, such diversity is not noise to average out but rather an informative signal. Even studies that previously reported this diversity and called for abandoning a strict dual gamma band view often ended up implicitly emphasizing high amplitude oscillatory events[8,12,13,27] (but see[15] for a criticism of independent multiple gamma views), which are spectrally more discrete (as the average spectrograms they dominate, Fig. 2d) but constitute only a rare minority of gamma elements. Here, beyond these studies, we find that also gamma elements with weaker oscillatory amplitude do carry task-related information, despite –or, eventually, precisely because of– their haphazard variability. Therefore, by taming complexity rather than ignoring it, we extend our investigation of oscillations-to-behavior interrelations to a broad majority of gamma elements, structurally ignored by previous approaches based on averages or "representative" patterns.

Next, we address the question of the causes underlying this gamma diversity. Some previous work hinted at their origin in non-linear collective dynamics of mesoscopic circuits (rather than supposing distinct cell types and connectivity motifs for distinct frequencies[14]). Our modeling analyses in Fig. 7a–d also suggest that gamma variability over the whole gamma range of frequencies is jointly generated through network-level mechanisms. In our model, it is the dynamics of random recurrent networks with balanced excitation and inhibition that inherently gives rise to complex ensembles of oscillatory events with fluctuating and diverse frequencies. This is achieved without the need of careful parameter tuning, as variability naturally arises as soon as the circuit is set to operate in a broad fringe in proximity but still below a transition to partially synchronized collective oscillations[28]. This contrasts with many previous models[29,30] which operated in more synchronous regimes (akin to the star working point in Fig. 7d): their narrowly tuned rhythms cannot indeed render the strong variability observed for individual gamma elements, although matching prescribed target frequencies based on average oscillatory patterns observed in vivo (but in our model the frequency of high-amplitude oscillatory events can also be tuned to be more "rad" or "lm" -like; cf. Supplementary Fig. 12).

Our modeling analyses also allow us to formulate hypotheses about the mechanisms through which gamma element fluctuations would acquire a dependence on behavior and learning. Indeed, in Fig. 7f, we show that the (lack of) firing of specific neurons can be reliably decoded from instantaneous gamma element variability. This implies that the fine details of this variability –the ones that machine learning classifiers learn to parse to extract behavior-related information– are partially shaped by the details of underlying neuronal firing dynamics. We thus believe that the information we extract from gamma elements is the blurred shadow of information conveyed by neuronal ensembles whose spiking eventually represents environment- and behavior-related aspects. Hence, the multiplicity of gamma frequencies would not indicate the existence of a multiplicity of channels for information routing, as in early interpretations of discrete gamma bands models[11]. Instead, it would reflect the irregularity of sparsely synchronized firing in randomly wired local networks, both resulting in large entropy of neuronal activity (Fig. 7d, right) and thus bandwidth for information coding. Note, that in regimes with higher synchrony, firing entropy would be severely reduced (see once again Fig. 7d, right), trimming, therefore, the amount of information that spiking could convey, relative to fringe-of-synchrony regimes associated with pervasive gamma element variability. Yet, high amplitude oscillatory events may still act as switches to transiently boost communication and enable the "push-forward" and "pull-back" of information encoded in rich spiking representations (as previously hypothesized[16]). This same theory[16] also predicts that low amplitude oscillatory transients would be scattered in phase, but high amplitude transients would phase-synchronize in a self-organized manner. Such prediction is well compliant with the observed properties of inputs received at the l-m and originating mostly from the entorhinal cortex, which are a mixture of low-power gamma elements scattered in phase with higher-power elements concentrated around a specific phase (Fig. 2, Supplementary Figs. 2 and 6). On the contrary, inputs received at the rad layer are scattered in phase at all power levels, suggesting that the generating populations in CA3 may be tuned in an even more asynchronous regime, only transiently ringing as an effect of filtering noise[31,32].

When decoding information from gamma elements, we would then be in reality capitalizing on indirect signatures of other codes, relying on cell ensemble firing[33], sequential activation of cell ensembles[34] or the dynamic selection of internal attractors or assemblies[35,36]. More in general, changing neuronal correlation and firing may translate into broadband deformations of the extracellular field power spectrum shape[37], thus explaining why information can be decoded even from oscillatory events at frequencies remote from spectral peaks.

Previous studies showed that increasing speed was associated with modulations of gamma amplitude, frequency or theta-phase, often with elaborate nonlinear relations, possibly dependent on learning[21,22]. The difficulty to identify relations between individual oscillatory features and speed may reflect the essentially synergistic, and thus conditional, nature of the mapping between gamma element variability and exploration behavior (Fig. 4e). Some aspects of the performance of our decoders may well hint at an influence of speed on the classifier decisions. For instance, it was easier to correctly identify presence in the target arm when speed was high, and in the reward field when speed was low. Similarly, gamma elements during fast movement in other arms tended to be misclassified as occurring in the target arm (cf. Supplementary Fig. 8b). Nevertheless, decoding performance remained well above chance level for all speed quartiles. Furthermore, the redundancies of individual gamma elements feature with speed (Fig. 4f) and the unique information about maze location conveyed by speed (Fig. 4e) both remained very low. Maze location affects the features of gamma elements in complex and synergistic ways, indicating that individual features do not exhibit a straightforward and direct relationship with behavioral information. Nonetheless, such a relationship does exist within the high-dimensional space encompassing all features, and our classifier effectively leverages this association for successful decoding purposes. This is particularly the case of the l-m gamma ensemble, whose synergistic mutual information levels where systematically larger than for the rad (cf. Supplementary Fig. 9), possibly reflecting richer and higher-dimensional encoding schemes by the activity of entorhinal cortex than of CA3 circuits. Nevertheless, there is no drastic change in prediction performance depending on the anatomical layer, indicating that all hippocampal layers received behaviorally-relevant information (likely through non-canonical hippocampal circuitry[38,39]) and that the information about location likely injected in the oscillatory activity by place cells firing do not provide the pyr with better positional correlates than gamma waves from other layers, this surplus of information being probably not visible at the rough spatial resolution level we use for our decoding. Indeed, our decoder does not finely predict animal position in the maze as some previous algorithms[40]. This is however not surprising, as our decoder operate on quite minimalistic inputs, parameterizing individual gamma elements at one time and at one recording location, while these previous algorithms exploited way more complex and higher-dimensional inputs. Furthermore, our aim was not to construct a performing decoder but to dismiss the hypothesis that gamma element variability is noise, and, for this less ambitious aim, the achieved performance is sufficient. However, it is likely that superior decoding, potentially allowing a finer identification of location beyond our rough subdivisions, becomes accessible when going beyond individual gamma elements to consider combinations of them (either temporal sequences or co-occurrences across layers).

Despite a substantial overlap in the frequency distributions of gamma elements at all layers, our cross-decodability analyses could yet reveal the existence of at least two types of gamma ensembles, corresponding to distinct blocks in the matrices of Fig. 6a and Supplementary Fig. 11b. Gamma elements recorded within the l-m (and the dentate gyrus) were modulated by maze location in very similar ways, but differently from channels within the more superficial rad. Interestingly, our cross-decodability approach exhibits a clear distinction between superficial and deep rad (Fig. 6a). Since cells located proximally in CA3 gave rise to collaterals that tended to terminate more superficially in the rad than did those arising from mid and distal levels of CA3[41], it is tempting to speculate that different CA3 inputs convey different information to CA1.

Remarkably, it is at the algorithmic level[26] of how behavior-related information is encoded that we can recover a clear distinction between

the rad and l-m layers that was not so evident at the level of frequency and phase distributions. Nevertheless, we provide a further confirmation that spatial location and navigation behavior are differently represented by the entorhinal cortex and CA3 circuits, serving as the input sources for functionally distinct gamma ensembles.

These representations are not fixed but evolve through the learning of the task, and more information about location can be decoded from late trials (Fig. 5b). Such improvement is paralleled by (and may be attributed to) an increase in the cross-decodability between layers, meaning that the nature of the representation is dynamically transformed. Decoders able to read sensory-related representations at layers receiving an entorhinal input become increasingly able to equally read model-based representations at layers innervated by CA3 (Fig. 6b). In other words, the grammar of sensory-related inputs, parsed by our decoders, becomes more and more compliant with the one of internal models. This result finds a natural interpretation within a predictive brain framework, since, with the learning of an internal model, the activity of sensory-processing regions shifts toward representing model-based inferences, beyond a passive encoding of external evidence[24].

Further, our results show that there is not a sharp distinction between naïve and expert types of location representations, but rather that these representations are smoothly adjusted through time as an effect of idiosyncratic experience. Our analyses indeed reveal an "arrow of time", with present decoders able to read out information from past gamma elements but not yet future ones (Fig. 5c, d). Each mouse has a different history of learning and, thus, potentially, a different way of coding rich individual behavior into differently organized but invariantly complex languages based on gamma ensembles (eventually, cross-classification between mice was not significant). A reduction in behavioral variability may contribute to the observed improvement in decoding accuracy during late trials compared to early ones. However, even in well-trained animals, there remains a considerable trial-to-trial variability in behavior. Despite this, a comprehensive characterization of these behavioral fluctuations is currently lacking. Future investigations may benefit from employing automated machine learning algorithms to categorize and analyze behavioral patterns[42,43], potentially enabling more sophisticated decoding approaches beyond basic maze location tracking. Note that, if gamma element variability is the manifestation of statistically rich firing dynamics, as hinted to by our modeling (cf. Fig. 7), our results imply that the organization and probability of occurrence of different cell assemblies is updated through learning in region-specific manners[44].

To conclude, hippocampal gamma activity is more diversified than a limited number of narrow frequency bands used by afferent generators at specific phases of the ongoing theta oscillation. At first sight, this variety may seem to threaten prominent views in which information from the two main hippocampal CA1 afferents is conditionally routed and disentangled at the neuronal level thanks to their distinct preferential frequency and theta-phase[13,27,45,46]. In these views, indeed it is a precise temporal and spectral separation of inputs that allows different structural pathways to mediate distinct cognitive functions[7,47]. Here, we show that such precise separation in frequency and phase most of the times does not occur. Yet, the diversity of gamma ensembles is not mere noise as it allows the successful decoding of behavior, is meaningfully coupled to both anatomy and learning and is likely to reflect the recruitment of sparsely synchronized and richly informative firing patterns, unavailable in more synchronized regimes. By emphasizing the relevance of low power events with "misbehaving" phase and frequency, usually discarded, our results suggest that system's function may rely on the self-organized coordination between noisy and weak oscillatory bursts[16] rather than on rigid architectures with precisely tuned oscillations.

## Methods

### Animals and surgery

**Subjects.** Five adults male CD1 mice (~3-month-old at the time of surgery) were housed in individual cages post-surgery, under a 12-h light/dark cycle (light at 8:00 A.M.). They had water and food ad libitum till the start of the habituation period; they were then water-restricted (2h-daily access, ~2 h after testing) for the entire duration of the experimental protocol. All experimental protocols agreed with the European Committee Council directive (2016/63/UE) regarding animal experimentation and were approved by the French Ministry of Research (APAFIS#20388-2019042517013497).

**Surgery.** Animals were anesthetized with isoflurane during the entire surgery. Linear silicon probes with either 16 or 32 channels (50 μm-spacing; A1x16-3mm-50-177-CM16LP or A1x32-6mm-50-177-CM32, Neuronexus, Ann Arbor, USA) were chronically implanted through the CA1-DG axis of the right dorsal hippocampus (AP: 2.06 and ML: 1.3 from bregma; DV: 1.7 from the dura). They were covered with DiI stain (Invitrogen Molecular probes, USA) before insertion. Two screws were positioned in the posterior and anterior portions of the skull, serving as ground and reference electrodes, respectively.

**Histological procedures.** The mice were perfused with 0.1 M PBS followed by 4% paraformaldehyde in PBS solution with added heparin (25 kUI). Brains were postfixed for 24 h in 4% paraformaldehyde before being cryoprotected in 20% sucrose solution for 48 h. They were then frozen in isopentane and sliced into 40-μm coronal sections. Implantation sites were visualized through a fluorescence microscope (Zeiss) thanks to the DiI stain.

### Behavioral apparatus and protocols

**Eight-arm radial maze.** The radial arm maze consisted in a central platform (52-cm diameter) from which eight identical arms (55 ×10 cm) expanded, separated by a 45-degree angle. Each arm was surrounded by a 3-cm high wall. A shallow circular recess at the end of each arm could hold the reward (75 μL of 5%-sucrose solution). The maze was situated 65 cm above the floor in a room displaying numerous distal visual cues that remained in position for the entire duration of the experiment. Mice were transferred from their home cage to the maze using an opaque box (start box: 20x10x15 cm).

**Habituation to the apparatus.** One week after surgery, mice were habituated to the experimental apparatus and the experimenter. A recording cable was plugged on a permanent basis to the head-mounted pre-amplifier so the mice could get used to its presence and weight. The animals were then handled by the experimenter for a few days before starting the habituation per se. This period consisted of transferring the mouse from its home cage to a single arm removed from the maze and placed elsewhere in the room. The mouse had to wait for 20 s in the start box (positioned at the entry of the arm) before the door opening. The aim was to reach the other end of the arm to consume the reward. This was repeated for three to five days, with five to eight trials a day (or until the mouse was not showing clear signs of anxiety). Mice were then exposed to the radial arm maze for two days during a daily 10-min trial in which the start box was positioned at the center of the maze and opened after 20 s. Every arm was reinforced only once per trial to promote exploration of all arms across both days. The inter-trial interval was five min, during which the apparatus was cleaned with 35% ethanol.

**Arm-to-Arm task.** In the Arm-to-Arm (ATA) task, mice must find the rewarded arm, the same across the 10 days of training (~24 h between sessions). The four daily trials start each from one of the four possible departure arms (two and three arms away from the target arm, both left and right; identical across sessions to ensure a constant distance to

the target) following a pseudo-random order to promote allocentric navigation. Hence, the animals wait in the start box positioned at the end of one arm for 20 s before opening of the door. It then has up to three min to find the rewarded arm otherwise the trial is stopped. The inter-trial interval was five min, during which the apparatus was cleaned with 35% ethanol. On the 11th day, a five-minute probe test is carried out to assess the animal's spatial reference memory: the mouse is released from a new departure arm (opposite to the target) and no reward is available. The mouse is considered to have learned the reward location if it either visited more often or spent more time in the target and its two adjacent arms than chance (proportion: 0.125).

## Electrophysiological recordings and analysis

**Recording and preprocessing.** The electrophysiological activity was recorded with an Intan recording controller (RHD Recording Controller, Intan Technologies, USA). The signals were amplified 200x, recorded whole-band (0.1–10 kHz), and digitized at 20 kHz. They were synchronized with a video system tracking the position of the animal at 20 Hz (Imetronic, France). The basic pre-processing of the LFPs included the removal of both slow variations and 50-Hz (and harmonics up to 200 Hz) electrical noise (Chronux Matlab toolbox[48]), artefact correction[49] and finally downsampling to 1 kHz.

**Anatomical localization of the electrodes.** Each electrode was assigned to an anatomical hippocampal layer depending on its distance from the hippocampal fissure along the estimated probe position in the histological slice. The theta power from each electrode was calculated by a group of complex Morlet wavelets (1–14 Hz by 1-Hz steps; 2-s duration; the number of cycles linearly dependent on frequency, between 2 and 4 cycles) on the LFPs filtered for theta range (4–12 Hz; zero-phase digital filtering using a finite impulse response filter of order = 256). The fissure was located at the peak of the Gaussian fit of the theta power curve, possibly between two electrodes.

**Signal decomposition.** For further analyses, instead of using a classic passband filter, we used an unsupervised, nonlinear, and non-stationary technique to isolate the dominant oscillations present in the LFPs in time, amplitude, and frequency: the Empirical Ensemble Mode Decomposition (EEMD[50]). The resultant components, termed Intrinsic Mode Functions (IMFs), can then be summed to recompose the original signal. Hence, to filter the LFPs in either theta (4–12 Hz) or gamma (30–250 Hz) frequency range, we summed the IMFs whose mean of the Hilbert-derived instantaneous frequency fell within the relevant range, thus obtaining a theta and a gamma composite LFP signals. For every trial, LFPs were decomposed independently for the period of actual navigation, that is, from when the animal is about to navigate in the maze (hence excluding start box or behavioral inactivity periods sometimes following the box opening) to up to 5 s following arrival to reward (or trial end if the animal did not find the water). Ten IMFs were requested, resulting from the average of 2000 iterations with added noise (input noise level of 0.3 except for some trials from two mice [mouse #3 and #4] needing 0.8 to satisfactorily alleviate mode mixing). To reduce confounds from potential theta harmonics, we started our gamma range at 30 Hz[8] unlike some previous reports of a lower bound at 25 Hz (see[15] for a recent review). Note that, to contrast our results with established methods (Supplementary Fig. 3), we also processed the signal using a finite impulse response filter combined with a zero-phase filtering for both theta and gamma bands or with an independent component analysis (KD-ICA algorithm within 'ICAofLFPs' Matlab toolbox[51]) instead of the EEMD decomposition.

**Theta cycles identification and selection.** Theta cycles were identified on the theta LFP composite from the closest channel to the hippocampal fissure. Using the fissure as a theta reference offers larger,

more defined theta cycles but implies an inverted theta phase compared to theta recorded within the CA1 pyramidal and oriens layers. Peaks in the signal were identified as the start/end of each candidate cycle. The trough was determined as the point with the lowest amplitude between two consecutive peaks, and the flanks, as the points at half-amplitude between the trough and these surrounding peaks. The theta phases (0–360°; peak = 0/360°) were obtained by linear interpolation within each quadrant formed by the starting peak-descending flank (0–90°), the descending flank-trough (90–180°), the trough-ascending flank (180–270°) and the ascending flank-next peak (270–360°). This waveform-derived phase determination is more respectful of the theta waves asymmetry than the one from the Hilbert transform[52,53] although we compared both methods (Supplementary Fig. 3). Note that EEMD-based composite signals are supposed to better respect the wave asymmetry than classic filters[12]. To be selected for analysis, the candidate theta cycles had to meet the following criteria[12]: a duration compatible with the theta frequency band (i.e., 83 to 250 ms) and a sufficient power (amplitude of the envelope of the theta LFP composite signal at the cycle start, mid and end points superior to the envelope of the 1–4 Hz infra-theta LFP composite signal). They further needed a coincident video sample to determine the animal position in the maze at that time.

**Amplitude of theta cycle-nested gamma.** To lessen volume-conducted activity, the amplitude of gamma oscillations was calculated on the current source-density (CSD) signal derived from gamma LFP composites as previously described for LFP[46]. CSD at a given time point $t$ was calculated as follows:

$$CSD(n,t) = \frac{-LFP(n-1,t) + 2 * LFP(n,t) - LFP(n+1,t)}{\Delta d^2} \quad (1)$$

where $LFP_{(n,t)}$ is the gamma LFP composite recorded at the electrode $n$, $LFP_{(n+1,t)}$ and $LFP_{(n-1,t)}$ are the gamma LFP composites from electrodes directly above and below, respectively, and $\Delta d$ is the distance (in mm) between contacts.

The continuous amplitude of the CSD signal, used as an instantaneous metric of power, was then obtained for each channel using complex Morlet wavelets convolution (0.5-s duration; from 15 to 200 Hz by 5-Hz steps and assessed by a number of cycles linearly dependent on the wavelet main frequency, between 6 and 20 cycles). The portion of this convolution corresponding to the time of each theta cycle was then isolated[12,13] and the CSD amplitude for each gamma frequency was averaged per theta phase (10° phase bins). Hence, the gamma spectral contents of each theta cycle were summarized in a 'snippet' (38 ×36 matrix: frequency x theta phase bin).

**Gamma bouts detection.** Within each individual theta-cycle, we extracted gamma elements as patches of locally higher gamma composite power in the CSD spectrogram. To identify these patches, we treated single-theta cycle spectrograms as color-scale images and binarized them, assigning to pixels with a gamma composite power larger or lower than a fixed threshold black or white color, respectively. We then used a standard flood-fill algorithm[54] to identify connected components within the binarized spectrogram image, each corresponding to a potential gamma patch. Since the number of connected components depend on the applied threshold, we decreased systematically the threshold starting from a value equal to the maximum power value within the original spectrogram. When reducing the threshold, more image pixels rise above threshold and the number of connected components tend to increase, apart from a few exceptions (see below). The scanning of decreasing threshold values stopped when a maximum (arbitrarily chosen) number of four connected patches (and thus gamma elements per theta cycle) was identified. Cases could arise in which the addition of new black pixels to the binarized image

caused patches disconnected at higher threshold values to finally merge. However, such patch fusion should be prevented, as the merged patch includes multiple and distinct power peaks. We thus added tracked record of the components' extensions immediately prior to merging, storing them as separated. Another special case needing ad hoc handling was the one of components located at the boundaries of the theta cycle and therefore potentially extending across two contiguous theta-cycle. To avoid double counting of a same component (detectable in both the cycles across which it is split), we thus parsed simultaneously neighboring theta cycles, to identify the complete extension of cross-cycle boundaries patches and count them only once (assigning them to the cycle over which the strongest amount of power was located).

After determining connected components segregated from the background and correcting for patch fusion and double counting, we then computed for each retained component the following gamma element features: gamma amplitude, frequency, and theta-phase of occurrence. Each pixel within a component was associated to a specific power, frequency, and phase triplet of values (respectively, the color, the vertical, and the horizontal coordinate within the single-theta cycle spectrogram image). The gamma element power was evaluated as the average power over all pixels within a connected component. The gamma element frequency and phase were then determined as the average among the frequencies and phases of the pixels within the component, weighted pixel-by-pixel by the pixel power.

We also compute the associated theta wave instantaneous frequency (inferred from cycle duration), amplitude (average voltage difference between the trough and the two adjacent peaks) and asymmetry (rise – decay ratio).

All gamma elements were appended to a list for each trial and channel, which was then filtered to exclude the top and lowest 1% amplitude gamma elements. Analogously, we excluded some gamma elements occurring in theta cycles coincident with unlikely large running speeds (> 100 cm/s).

## Dimensionally reduced representations of gamma elements

We used a standard t-Stochastic Neighborhood Embedding (t-SNE) algorithm[19] to create bidimensional representations of the diversity of gamma elements. This algorithm guarantees that distance inter-relations between data-points in the source high-dimensional space are preserved as much as possible in the target bidimensional space representation. The projection was learned for all gamma elements simultaneously (all layers and trials), and then different groups of elements could be shown in different panels (see. Figure 1f, g and Supplementary Fig. 5) filtering the same common and frozen projection. We used standard hyperparameters (perplexity = 30, no exaggeration) with an approximated Barnes-Hut algorithm. We used an Euclidean distance metric except for the theta phase of gamma appearance where circular distance was used.

## Maze location classifier training

**General classification approach.** We used first a supervised classification approach to predict rough location within the maze based on an input vector parameterizing individual gamma elements. The input was given by a six-dimensional vector including general information about the theta cycle (theta-composite amplitude, frequency and asymmetry) and specific features of the considered element (gamma element power, frequency, and theta phase of occurrence), computed as described in previous sections. The output was a categorical label, referring to a subdivision of the maze in four sections: "Reward RF" (end-field of target arm where reward was delivered); "target arm" (including the arm leading to the reward location and the outer area of approach to this same arm, "); "other RFs" (arm end field other than the reward field); and, finally, "other locations" (including all areas not including in the previous

subdivisions, i.e. maze center and generic maze arms not leading to reward).

As multi-class classifiers, we used boosted ensembles of classification trees, limiting the maximum number of decision splits in a tree to 500, and the number of learners in an ensemble to 500 trees. Tree ensembles were fitted using the RUSBoost algorithm[55], with a slow learning rate of 0.01, to alleviate the problem of output class unbalance (as some classes, as "target arm" or "Reward RF" are under-represented relatively to others). In this algorithm, random under-sampling is applied to training sets to guarantee that each class is represented by close numbers of samples, providing simultaneously the capability to learn rare classes and protection against biases due to variations across different conditions (e.g. early vs late trials) of the fractions of samples per class.

**Classification performance.** Classification performance was evaluated both in terms of resubstitution error (error on same data samples used for training) and generalization error (error on data samples not used for training), estimated via 4-fold cross-validation. In generating the random partitions into training and testing sets of elements, beyond output class balancing, we took care to use for testing gamma elements measured in theta cycles not included in the training set, thus conferring protection against overfitting. The list of theta cycles (and gamma elements therein) available for a selection of training and testing pools corresponded to the total list of elements retained for a channel in a mouse, over all the trials (unless otherwise specified, see next section on cross-classification). Different classifiers were trained independently for each different channel. Figure 2d–f and Supplementary Fig. 8 report average performances over all CA1 channels, as classification performance was shown to have only weak dependency from the layer (cf. Figure 2c and Supplementary Fig. 7). Generally, unless otherwise specified, classification performances (and misclassification rates) are expressed in terms of correct (incorrect) classification fraction, evaluated over all available gamma elements (cross-validation ensuring that prediction on an element was performed in terms of classifiers not having seen this element during training). Despite cross-validated training had access to the whole list of gamma elements extracted from a channel, after training, we could also evaluate classification performance on subsets of gamma elements to assess whether the probability of correct classification depended on various features of the gamma elements fed as input. We thus separated gamma elements according to them belonging to different quartiles of the distribution of different features (from Q4, with top values, to Q1, with the lowest values): the six features of the gamma element descriptive vector (gamma power, frequency and phase, theta amplitude, frequency and asymmetry), as well as motion speed (averaged over the time range of the considered theta cycle); and computed fractions of correct classification separately over each distribution quartile.

**Classification with alternative input features.** We also trained classifiers using alternative reduced sets of input features. Instead of using the full six-dimensional descriptive vectors of gamma elements as input vector (as in the "theta + gamma" classifiers just described), the "theta-only" (or "gamma-only") classifiers were trained just in terms of the three theta (or gamma) features entries.

**Classification of alternative arm and in probe trials.** We also used an alternative set of output labels in which the "Target arm" section of the maze was merged with the "other locations" section, but in which an "Alternative arm" was considered instead as a separate section, with the same extension of the Target arm zone (arm plus outer arm approach zone) but including an arm different from the one leading to reward. This alternative arm was chosen to be opposite to the target one. In this alternative zone labeling, the "reward RF" zone was left

unchanged, i.e. it still included the reward location (and was therefore not contiguous to the "Alternative arm" zone). Classifiers were trained on this new labeling in a completely independent way from the classifiers trained on the original labels.

Probe trials were not used for training, but location prediction was performed using classifiers trained over control trials, with the ordinary output zone labeling (i.e., including the "Target arm" and not the "alternative arm" zone).

## Cross-classification

**General approach.** Once trained, a classifier serves as an implicit model of the distribution of gamma ensembles in relation to behavior. Changes of the ensembles-to-behavior relation across conditions or channels can be studied using a cross-classification approach, in which classifiers trained on a sample are evaluated on a different sample. Preserved or decreased performance levels will then denote, respectively, similarity or dissimilarity or relation to behavior.

**Cross-classification through learning.** To study evolution of the ensemble-to-behavior mapping across task learning by the mice, we selected gamma ensembles over subsets of trials only. Specifically, we sorted all trials available from the earliest to the latest and compiled a table of how many gamma elements each trial provided on average over all channels. We then defined two "early" and "late" trials ranges, including trials with ordinal numbers respectively smaller and larger or equal than a pivot trial number. This pivot trial number was chosen such that the cumulative sums of gamma ensemble counts per trial over the early and late ranges were as close as possible between them. The early and late trial range specifications were therefore adapted to the actual behavioral history of each mouse. Furthermore, the early and late trial ranges usually included unequal numbers of trials, as maze exploration is faster in later than in earlier trials and, consequently, individual late trials usually contribute smaller counts of gamma ensembles.

We then adopted a finer subdivision of trials when constructing the cross-classification matrices of Fig. 3c and Supplementary Fig. 11a. Once again, we ordered trials and grouped them into smaller window ranges such that the cumulative sum of gamma ensemble counts for the trials included in each window was as close as possible to 3000 elements. Every window included all trials with ordinal numbers between the ones of a start and stop trials. Windows could have an overlap, but two consecutive windows could not have the same start and stop trials. Different windows generally included different numbers of trials, with windows at earlier times being generally narrower than windows at later times.

Classifiers were then trained over just the early or late range of trials, or, yet, just trials within a specific learning window, using the same class-balanced, cross-validated approach described in the previous section. The partial datasets were randomly downsampled to exactly include the same number of elements (as the numbers of elements provided by early and late trials ranges or by different windows were close between them, but not identical). Although cross-validation was still used in training, it could not be systematically used in evaluating cross-classification performances, as the original and checking datasets did not include the same theta cycles and partitions generated for the one was thus invalid for the other. Therefore, in the cross-classification performance matrices of Fig. 3c and Supplementary Fig. 11a, we reported average resubstitution error along the diagonal and, in off-diagonal entries, direct average performance on the considered checking dataset.

The improvement of decodability in late relatively to early trials (Fig. 3b) was evaluated, for each mouse, as the relative percent difference between cross-validated performance, averaged over all classes, of classifiers trained just on late or early trials (for a representative channel in the middle of rad or of the l-m layers). The performance

asymmetry in classifying past vs future trials (Fig. 3d) was evaluated as a relative percent difference between the averages of the upper and lower triangular parts of the cross-classification matrices of Fig. 3c and Supplementary Fig. 11a.

**Cross-classification through channels.** Classifiers trained on a channel were used to extract location based on gamma elements of another channel. We computed cross-classification performance across channels based on the whole set of available trials (Fig. 3e and Supplementary Fig. 11b) and also based on just trials in the early and late ranges. For a better comparison with cross-classification analyses across trials we once again computed cross-classification performances and relative percent variations in terms of resubstitution and direct checking errors. Cross-classifiability between rad and l-m layers was evaluated averaging cross-classification matrix entries in blocks delimited by channel ranges matching the different layers. Note that some uncertainty exists at layer edges, as electrodes could slightly move from one day to the next, causing some of them to transit above or below the depth delimiting two layers. A channel was thus included in the block average only if it belonged to a specific layer in at least three quarters of the trials used to build the classifier.

## Information-theoretical analyses of element features to behavior relation

**Mutual Information between pairs of features and maze location.** To study the nature of the relation existing between different descriptive features of the gamma element and maze location, we complemented decoding by machine-learning classifiers with information theory analyses[56] and computed mutual information between pairs of input variables and simultaneously visited maze location. We used a rough estimation of the probability distributions of input variables, quantizing them into four unequal bins, matching the distribution quartile limits. By replacing feature values by their quartile label in the feature distribution, we then automatically maximized single variable entropies, as entropy for discretized variables is maximal for uniform distributions. Output labels were already categorical and in a number of four, corresponding to the four rough maze sections previously described (Reward RF, Target Arm, Other RF, Other locations). For each pair of quantized input features $f$ and $g$ and output maze location labels $L$, we computed over the list of all gamma elements for representative channels in rad and l-m layers the joint normalized frequency histogram $P(f,g,L)$ and, out of it, the total mutual information that the pair of inputs $(f,g)$ carries about the output $L$:

$$I(f,g;L) = \sum_{f,g,L} P(f,g,L) \log_2 \frac{P(f,g,L)}{P(f,g)P(L)} \qquad (2)$$

normalized by the total entropy $H(L) = -\sum_L P(L)\log_2 P(L)$ of the output variable (to quantify the fraction of location information carried by the pair of input features).

**Partial Information Decomposition.** We then decomposed this total mutual information using the Partial Information Decomposition (PID) framework[23] into: unique fractions of information, i.e. information that $f$ (or $g$) carry about $L$ but that $g$ (or $f$) don't carry; a redundant fraction of information, i.e. information that both $f$ and $g$ carry about $L$; and a synergistic fraction of information, i.e. information that neither $f$ or $g$ alone carry about $L$ but that they carry when jointly considered. We evaluate the synergistic information of $f$ and $g$ relative to $L$ as:

$$Syn(f,g;L) = I(f,g;L) - I(f;L) - I(g;L) + Red(f,g;L) \qquad (3)$$

where $Red(f,g;L)$ is the redundant fraction of information and must be added back once because twice removed from the total mutual information $I(f,g;L)$ when subtracting $I(f;L)$ and $I(g;L)$, mutual

information of $L$ which just $f$ or just $g$. To estimate redundancy, we use the so-called Minimal Mutual Information ansatz, under which the redundant information fraction is made to correspond exactly to the minimum between the two individual mutual information terms, i.e. $\text{Red}(f, g ; L) = \min[\text{I}(f ; L), \text{I}(g ; L)]$. In this way, the unique information carried by the least informative of the two variables (say, $g$) is set to zero, and the remaining difference equated to the unique information carried by the most informative variable, i.e. $\text{Unique}(f ; L) = \text{I}(f ; L) - \text{I}(g ; L)$. Unique, redundant and synergistic fractions of the total mutual information can also be normalized by the entropy of the stimulus. Figure 2g shows average total mutual information with location and decomposed fractions, averaged over all pairs of input features including a specific reference feature. Besides the six features describing gamma elements we also considered pairs of inputs including motion speed $V$ as input variable, discretized in a quantile-based way as the other features. We also analyzed separately the four gamma elements extracted out of each theta cycle, ranking them in decreasing order of power, to reveal whether the informative content of elements concentrated on the strongest power elements or was uniform across stronger or weaker gamma power elements. Details about the decomposition for specific pairs of features are shown in Supplementary Fig. 9.

**Redundancy with speed.** The general dependency of gamma element features on speed could be assessed by the redundancy between discretized gamma element features $f$ and the speed variable $V$, i.e. $\text{Red}(f ; V) = \text{I}(f ; V)$. Such redundancy was then normalized by the entropy of $f$, to quantify the fraction of information about the variability of $f$ explained by the variability of $V$ (cf. Figure 2f).

### Computational model of gamma elements generation
**Model definition and parameters.** We considered a network composed of $n = 2000$ quadratic integrate and fire (QIF) neurons[25], 80% of them excitatory (E) and 20% inhibitory (I). The membrane potential $\nu^E_j (\nu^I_j)$ of each excitatory (inhibitory) neuron $j$ obeyed the following differential equations:

$$\tau^E_m \dot{\nu}^E_j = \left(\nu^E_j\right)^2 + I_\theta(t) + I^E_j$$
$$+ 2\tau^E_m \left[ g^{EE} \sum_{l:t^{(n)}_l < t} \varepsilon^{EE}_{jl} \delta\left(t - t^{(n)}_l\right) - g^{EI} \sum_{l:t^{(m)}_k < t} \varepsilon^{EI}_{jk} \delta\left(t - t^{(m)}_k\right) \right], \quad (4)$$

$$\tau^I_m \dot{\nu}^I_j = \left(\nu^I_j\right)^2 + I_\theta(t) + I^I_j$$
$$+ 2\tau^I_m \left[ g^{IE} \sum_{l:t^{(n)}_l < t} \varepsilon^{IE}_{jl} \delta\left(t - t^{(n)}_l\right) - g^{EI} \sum_{l:t^{(m)}_k < t} \varepsilon^{II}_{jk} \delta\left(t - t^{(m)}_k\right) \right], \quad (5)$$

where $\tau^E_m = 10\text{ms}$ ($\tau^I_m = 4.5\text{ms}$) is the excitatory (inhibitory) membrane time constant and $I^E_j (I^I_j)$ the neuronal excitability encompassing single neuron characteristics as well as synaptic drives originating from other neural regions and acting on the excitatory (inhibitory) neuron $j$. The input term $I_\theta(t)$ is a forcing current periodically modulated at a θ-like frequency of 10 Hz and $g^{\alpha\beta}$ the synaptic coupling strength between a post-synaptic neuron $s$ in population $\beta$ and pre-synaptic neurons in population $\alpha$, with [α,β] being either E (excitatory) or I (inhibitory). The connectivity matrix elements $\varepsilon^{\alpha\beta}_{jl}$ are equal to one (zero) if a connection from a pre-synaptic neuron $l$ of population $\beta$ towards a post-synaptic neuron $j$ of population $\alpha$, exists (or not). Furthermore, $k^{\alpha\beta}_j = \sum_l \varepsilon^{\alpha\beta}_{jl}$ is the number of pre-synaptic neurons in population $\beta$ connected to a neuron $j$ in population $\alpha$, or, in other terms, its in-degree restricted to population $\beta$. The emission of the $n$-th spike emitted by neuron $l$ of population $\alpha$ occurs at time $t^{(n)}_l$ whenever the

membrane potential $\nu^\alpha$ crosses threshold for firing, while the reset mechanism is modeled by resetting $\nu^\alpha$ to a rest value, immediately after the spike emission (see[57] for details on threshold and reset in QIF neuron model). For the sake of simplicity, we assumed synapses to be fast and synaptic transmission instantaneous, therefore the post-synaptic potentials were modeled as δ-pulses without any delayed activity. Connectivity within the E and I populations was random and quenched, with in-degrees $k^{\alpha\alpha}$ distributed according to a Gaussian distribution with mean $K^{\alpha\alpha}$ and with a standard deviation $\Delta^{\alpha\alpha}$, this latter parameter measuring the level of structural heterogeneity in each population. We chose here to set $K^{EE} = K^{II} \equiv K$, providing a common scale for the strength of local connectivity in the model. As a further simplification (suitable for potential mean-field reduction not explored in this study), we then assumed that the E and I populations are globally cross-coupled, i.e. $\varepsilon^{\alpha\beta}_{jl} = 1$, for any $j,l$ if $\alpha \neq \beta$. The neuronal excitabilities $I^\alpha_j$ were distributed according to a Gaussian distribution with mean $I^\alpha_0$ and standard deviation $D^\alpha$. The DC currents and the synaptic coupling were rescaled with the median in-degree as $I^\alpha = I^\alpha_0 \sqrt{K}$ and $g^{\alpha\beta} = g^{\alpha\beta}_0 / \sqrt{K}$ to obtain a self-sustained balanced dynamics for $K \to \infty$[28,58,59]. The structural heterogeneity parameters were rescaled as $\Delta^{\alpha\alpha} = \Delta^{\alpha\alpha}_0 \sqrt{K}$ in analogy to Erdos-Renyi networks[28]. We employed, unless stated otherwise, the following values of the parameters: $I^E_0 = 0.3$; $I^I_0 = 0.25$; $D^E = 0.1 \cdot I^E_0$; $D^I = I^I_0;K = 20$; $\Delta^{EE} = 2 \cdot K$; $\Delta^{II} = 0.2 \cdot K$; $g^{EE}_0 = 0.27$; $g^{II}_0 = 1.44 \cdot g^{EI}_0 = g^{IE}_0 = 0.01$. The θ-forcing was assumed to be perfectly sinusoidal, as $I_\theta(t) = A\sqrt{K} \cos(2\pi\nu t)$, with $\nu = 10\text{Hz}$ and $A = 0.042$.

**Simulated local field potential.** The Local Field Potential was modeled as LFP$= (|I_A| + |I_G|)$, which is the sum of the absolute values of AMPA and GABA currents impinging on pyramidal cells, following[60]. The global currents $I_A$ and $I_G$ were the linear sum of contributions induced by single pre-synaptic spikes, each represented as a combination of two exponentially decaying functions. This representation can be obtained using auxiliary variables $x_{Aj}, x_{Gj}$. The time evolution of AMPA and GABA-type currents of neuron j were thus described by the following ordinary differential equations:

$$\tau_{dA} \frac{dI_{Aj}}{dt} = -I_{Aj} + x_{Aj} \quad (6)$$

$$\tau_{dG} \frac{dI_{Gj}}{dt} = -I_{Gj} + x_{Gj} \quad (7)$$

$$\tau_{rA} \frac{dx_{Aj}}{dt} = -x_{Aj} + 2\tau^E_m g^{EE} \sum_{l:t^{(n)}_l < t} \varepsilon^{EE}_{jl} \delta\left(t - t^{(n)}_l\right) \quad (8)$$

$$\tau_{rG} \frac{dx_{Gj}}{dt} = -x_{Gj} + 2\tau^E_m g^{EI} \sum_{l:t^{(m)}_k < t} \varepsilon^{EI}_{jk} \delta\left(t - t^{(m)}_k\right) \quad (9)$$

where $\tau_{dA}(\tau_{dG})$ and $\tau_{rA}(\tau_{rG})$ are respectively the decay and rise time of the AMPA-type (GABA-type) synaptic currents. Always following[60], we high-pass filtered the obtained model LFP signal at 1 Hz with a 4th order Butterworth filter and employed $\tau_{rA} = 0.4\text{ms}$; $\tau_{rG} = 0.25\text{ms}$; $\tau_{dA} = 2\text{ms};\tau_{dG} = 5\text{ms}$.

**Numerical simulations.** Numerical simulations of the model were performed with a standard Euler integration scheme with time step $\delta t = 0.001$ ms. Since all disorder in connectivity and conductance is quenched, a deterministic integration scheme can be used as in ref. 57. Simulations were performed scanning a range of $K$ and $g^{IE}_0$ values to explore different dynamical regimes (cf. Fig. 4 and Supplementary Figs. 12 and 13).

**Indicators of dynamic regime.** After the generation of synthetic LFPs, gamma elements could be extracted from them following the same procedures as for real LFP and CSD signals. We also computed an indicator of power distribution across frequencies, computing spectral entropy. To do so, the power spectrum $P(f)$ of simulated time-series was computed and over a range between $f_{min} = 25$ Hz and $f_{max} = 125$ Hz sampled at $df = 0.1$ Hz and normalized to provide a density functional. We then evaluated spectral Entropy as the quantity $E = -\frac{1}{E_{max}} \sum_f P(f) \cdot \log_2(P(f))$, where $E_{max} = \log_2(M)$ and $M = (f_{max} - f_{min})/df$. Higher values of $E$ correspond to higher dispersion of power across gamma frequencies.

We also estimated the ratio between the amounts of high- and low gamma power, estimating the total power in the low (25–50 Hz) gamma band $P_{low}$, and the total power in the high (50–100 Hz) gamma band $P_{high}$. The ratio $r_{gamma} = (P_{low} - P_{high})/(P_{high} + P_{low})$ indicated whether the LFP is dominated by high gamma (positive $r_{gamma}$), by low gamma (negative $r_{gamma}$), or by an equilibrated mix of the two ($r_{gamma}$ close to zero).

As variance in time domain is proportional to power magnitude in spectral domain, we computed standard deviation of the mean membrane potentials across inhibitory neurons as a measure of the amplitude of generated gamma oscillations.

Finally, we evaluated the average entropy of spike trains. To do so, we would ideally resort to a binning procedure in which every bin match a cycle of ongoing gamma oscillations and the binned spike train is binarized into a sequence where a symbol "1" or "0" are added whenever the bin does (not) contain at least a spike from the considered neuron. The entropy of the neuron spike train would then be H = $-p \cdot \log_2(p) - (1 - p) \cdot \log_2(1 - p)$, where $p$ is the probability that a bin is assigned the symbol "1". Pragmatically, given the large variability of gamma cycle lengths and the difficulty to segment them, we evaluated an approximated expression of $p$, estimated as the ratio between the average firing rate of the considered neuron and the average period (i.e. inverse of frequency) of gamma oscillations (cf. Supplementary Fig. 13a) and plugged then this estimate for $p$ into the above formula for H. As the firing rate of E and I neurons are rather different and would require different bins, making difficult comparisons across values obtained for different bin choices, we decided to focus on E neurons only for entropy analyses.

The values of spectral and spike train entropy, high/low gamma power ratio and standard deviation of potential were computed for simulations of models with different connectivity to study the dependency on them of the obtained dynamical regimes.

**Neuronal firing classifier training**
We also used a similar RUSBoost tree ensemble classifier design to predict not the location in the maze but the presence or absence of a spike emitted by a given neuron $i$ in a window of width 100 ms centered on the time of occurrence of an input gamma element. Specifically, we checked for the presence of at least a "1" in a -100 ms-long chunk of the binarized spike trains used for entropy computation, centered on the bin including the gamma element time. We thus focused on decoding, exclusively on E neurons, as the bin size used for binarization has been optimized for the E firing rate. Gamma elements were parameterized for maze location classifiers. Training and testing data came from computational model simulations, as in this case the exact firing state of each neuron in the network and its relation with the extracted gamma elements are known with precision. We used the same hyperparameters and cross-validated training scheme for classifying maze location. A different tree ensemble was trained for the prediction of the firing of each neuron. Performance was evaluated in terms of the True and False Positive (correctly or wrongly detected "1", i.e. spike presence, TP and FP) and True and False Negatives (correctly or wrongly detected "0", i.e. spike absence, TN and FN), in terms of

which we also evaluated the Precision, i.e. TP/(TP + FP) and Recall, i.e. TP/(TP + FN).

The performance was heterogeneous across neurons and two groups could be distinguished as evident from the bimodality of the TP and TN scores joint distributions (cf. Supplementary Fig. 13c). We call these two peaks the "undecodable" and "decodable" subgroups of neurons in the model and we empirically separate them by segmenting the "decodable peak with the conditions, TP > 0.6 and TN > 0.6, as justified by the found joint distribution shape. We statistically compared firing rate, excitability $I_O$ and in-degree $k$ between the two groups but found no statistically significant difference (Kruskal-Wallis test of medians). However, the two groups could be statistically distinguished by their phase concentration defined as:

$$\varphi = \left\| \frac{1}{K} \sum_{k=1}^{K} e^{-i\phi_k} \right\| \tag{10}$$

where $\varphi_k$ is the phase of the ongoing theta oscillation at which the $k$-th spike of the considered neuron is emitted and ‖•‖ denotes modulus of the complex number.

**Statistical analysis**
All statistics were performed using either built-in Matlab (R2021a) functions, Matlab toolboxes, or Statistica 13.

Electrophysiological data was analyzed on all 40 trials per animal except for mice mouse #3 (missing trial 15) and mouse# 4 (missing trials 21–24) due to technical issues with the electrophysiological recording.

**Behavior.** Average latency to reach the reward during the learning phase of the task (i.e., days 1–10) was analyzed by a non-parametric Friedman ANOVA (within-factor: days). For the probe test, we carried out a repeated measure ANOVA (within-factor: arms) on the ratio of a number of visits in each arm compared to the chance level (0.125). Post-hoc tests were used when appropriate. For all analyses, the significance threshold was 0.05.

**Distributions of gamma bouts features.** The probability density function (pdf) of each gamma feature (amplitude, frequency, and theta-phase) was established across all trials, per electrode. As the pdfs of gamma frequency often displayed a wide range whatever the channel and the animal, we restricted most of our analyses to one representative channel per anatomical layer: the channel displaying the pdf the most consistent with expectations from the dual-gamma band literature, usually based only on very strong gamma episodes (here: the strongest 5% gamma bouts). Hence, we favored channels showing a dominance of lower gamma frequencies in the rad, and faster gamma frequencies in the l-m. The mean pdf of each layer was calculated across mice, before generating an artificial sample of the relevant gamma feature matching this pdf ($n = 10000$; random sample with replacement). For the frequency, the statistical difference in the distribution from each pair of layers was evaluated using a bootstrap method (2000 repetitions using 1500 subsamples) on the Kullback-Leibler divergence[61] so that to be significant, the mean divergence between separate layers had to be greater than the upper 95% confidence interval on the mean divergence between shuffled layers. In addition, both the modes of the individual mouse frequency pdfs and the ratio between $gamma_M$ and $gamma_S$ were compared across layers using one-way ANOVA or non-parametric Kruskal-Wallis tests on the between-factor 'layer', with post-hoc

comparisons. The ratio was defined as:

$$\text{ratio} = \left(\frac{\text{gamma}_M - \text{gamma}_S}{\text{gamma}_M + \text{gamma}_S}\right) \qquad (11)$$

with gamma$_S$ and gamma$_M$ being the sum of probabilities for the individual mouse frequency range centered on the frequency modes between 25–50 Hz and 60–100 Hz, respectively, and whose probability is ≥ 50% of this mode.

For the phase, all statistical analyses were carried out using the 'circStat' Matlab toolbox[62]. First, the statistical difference between artificial distributions from each pair of layers, generated as for the frequency mean pdf, was assessed using the Kuiper two-sample test (note that very similar results were obtained using the circular Watson's U2 test with 1000 permutations). Second, their mean phases were compared by pairs of layers using the Watson-Williams test for circular means after checking the non-uniformity of these distributions (omnibus test). This latter analysis was also done on the individual mouse distributions ($n = 5$) to compare the grand mean phase across layers. All the above analyses were performed iteratively on distributions containing a varying range of data, from all data ($0^{th}$ percentile: no further data selection) to the $95^{th}$ percentile of the maximum amplitude (i.e., only the gamma bouts with the 5% strongest amplitude), by steps of five percentiles. Percentiles were calculated for each trial and electrode before pooling the bouts from all trials per electrode.

**Classification, cross-classification and mutual information.** For single mouse performance levels (fraction of classification correct and confusion matrices), as well as for information-theoretical quantities we evaluated 95% confidence intervals using a bootstrap with replacement approach (1000 replicas) over the lists of gamma elements retained for inclusion in each of the analyses. When comparing multi-mouse samples of performance metrics or testing their significance against a threshold (as in the boxplots of Figs. 2–3 and Supplementary Fig. 8), we used t-test (two-tailed for inter-sample comparisons and one-tailed for comparison of single samples against a chance-level or zero threshold). We report uncorrected p-values in captions and text, however significance, unless specified otherwise, is assessed using Bonferroni correction for multiple comparisons (\*, \*\*, \*\*\* denote corrected p-values smaller, respectively, than 0.05, 0.01, 0.001; symbols in brackets indicate significance only prior to multiple comparisons correction; when significant deviations in both directions above or below chance level occur, we use upward ↑ or downward ↓ symbols instead of \*'s). Boxes in the boxplot mark the inter-quartile range (IQR), the horizontal line sample mean μ, the whiskers μ ± 2\*σ where σ is sample standard deviation.

**Reporting summary**
Further information on research design is available in the Nature Portfolio Reporting Summary linked to this article.

## Data availability
We have made available a large set of raw and derived data in the public repository Zenodo, as loadable MathWorks Matlab (https://www.mathworks.com/products/matlab.html) workspace files. These online resources are accessible at the link https://doi.org/10.5281/zenodo.10181305. All other data will be made available upon reasonable request.

## Code availability
We made available on the same public repository as for data, three Mathworks Matlab Live Notebooks and one Python Jupyter Notebooks allowing the reproduction of all the key analyses of this study (gamma element extraction, classifier training and comparison, information

theory analyses of feature relevance, computational model simulation). Note that the needed data for running the included examples can be directly loaded from these same live notebooks. These online resources are accessible once again at the link https://doi.org/10.5281/zenodo.10181305.

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

## Acknowledgements

This work was supported by CNRS, Inserm, Université de Strasbourg, Aix-Marseille Université and grants from the University of Strasbourg Institute for Advanced Studies (USIAS 2020-044) to DB, from the Interdisciplinary Thematic Institute NeuroStra, as part of the ITI 2021-2028 program of the University of Strasbourg, CNRS and Inserm (IdEx Unistra ANR-10-IDEX-0002 under the framework of the French Program "Investments for the Future") to DB and RG and from the Agence Nationale de la Recherche: ANR ERMUNDY (ANR-18-CE37-0014) to DB, MdV and AT; ANR DG-Goal (ANR-17-CE37-0002) to RG; ANR HippoComp

(ANR-21-CE37-0011) to RG and DB, Labex MME-DII (USIAS) to AT. The authors wish to thank Pr. Jesse Jackson for critical reading of the manuscript and the Buzsaki's lab for publicly sharing their data.

## Author contributions

V.D. and R.G. performed the experiments and analyzed the data, M.D.V. and A.T. built the model, D.B. analyzed the data and built the classifier. R.G. and D.B. conceived the experimental and analytical design; all authors wrote the article.

## Competing interests

The authors declare no competing interests.
