## [Peer Review File · Nature Communications]

Gamma oscillatory complexity convey behavioral information in hippocampal networksEditorial Note: This manuscript has been previously reviewed at another journal that is not operating a transparent peer review scheme. This document only contains reviewer comments and rebuttal letters for versions considered at *Nature Communications*.

REVIEWERS' COMMENTS

Reviewer #2 (Remarks to the Author):

The authors added several new analyses to the ms and clarified the text regarding my questions.

I am OK with the present version.

Reviewer #3 (Remarks to the Author):

I have two remaining issues related to reviewer one:

I think the labeling of Figure 2d-f is still incorrect. It shows 0.2-0.8 % classification. I assume they mean 20-80%, otherwise, I would have to agree with referee 1 that the classification is very poor.

Second:

Question

"The raster plots shown for excitatory neurons in Figure 2b do not resemble the sparse firing patterns of excitatory neurons in the hippocampus of behaving rodents."

Answer

The patterns of firing are rather sparse, as quantified now in the new version of Figure 4 where we also compute spike train entropy in bins of the size of an average gamma cycle. So, neurons fire on average every ~4-5 gamma cycles in a quite stochastic manner, as quantified by an average CV of ~0.9 (near Poisson).

I agree with the original comment of the reviewer - firing at every 4-5 gamma cycles leads to >10Hz firing, which is much higher than the average firing of pyramidal populations (<1Hz). This discrepancy between real neuronal networks and the model should be acknowledged in the text.

Reviewer #2

The authors added several new analyses to the ms and clarified the text regarding my questions.

I am OK with the present version.

We wish to thank the reviewer for his work on reviewing our manuscript and we're glad we could answer all of her/his comments.

Reviewer #3

I have two remaining issues related to reviewer one:

I think the labeling of Figure 2d-f is still incorrect. It shows 0.2-0.8 % classification. I assume they mean 20-80%, otherwise, I would have to agree with referee 1 that the classification is very poor.

We thank the reviewer for spotting this mistake. The reviewer is right, the axis is between 20 and 80% of correct decoding. All the figures have been modified accordingly.

Second:

Question

"The raster plots shown for excitatory neurons in Figure 2b do not resemble the sparse firing patterns of excitatory neurons in the hippocampus of behaving rodents."

Answer

The patterns of firing are rather sparse, as quantified now in the new version of Figure 4 where we also compute spike train entropy in bins of the size of an average gamma cycle. So, neurons fire on average every ~4-5 gamma cycles in a quite stochastic manner, as quantified by an average CV of ~0.9 (near Poisson).

I agree with the original comment of the reviewer - firing at every 4-5 gamma cycles leads to >10Hz firing, which is much higher than the average firing of pyramidal populations (<1Hz). This discrepancy between real neuronal networks and the model should be acknowledged in the text.

We thank the reviewer for this comment. We now have added a comment in the results section of our manuscript: "note however that in our model, firing rate of excitatory neurons is higher than the one recorded in the hippocampus or entorhinal cortex of behaving rodents" (Line 371-372).